# Mitochondria supply ATP to the ER through a mechanism antagonized by cytosolic Ca$^{2+}$

**Jing Yong[1], Helmut Bischof[2], Sandra Burgstaller[2], Marina Siirin[1], Anne Murphy[3], Roland Malli[2], Randal J Kaufman[1,3]***

[1]Degenerative Diseases Program, SBP Medical Discovery Institute, La Jolla, United States; [2]Molecular Biology and Biochemistry, Gottfried Schatz Research Center, Medical University of Graz, Graz, Austria; [3]Department of Pharmacology, University of California, San Diego, La Jolla, United States

**Abstract** The endoplasmic reticulum (**ER**) imports ATP and uses energy from ATP hydrolysis for protein folding and trafficking. However, little is known about how this vital ATP transport occurs across the ER membrane. Here, using three commonly used cell lines (CHO, INS1 and HeLa), we report that ATP enters the ER lumen through a cytosolic Ca$^{2+}$-antagonized mechanism, or *CaATiER* (**Ca$^{2+}$-A**ntagonized **T**ransport **i**nto **ER**). Significantly, we show that mitochondria supply ATP to the ER and a SERCA-dependent Ca$^{2+}$ gradient across the ER membrane is necessary for ATP transport into the ER, through SLC35B1/AXER. We propose that under physiological conditions, increases in cytosolic Ca$^{2+}$ inhibit ATP import into the ER lumen to limit ER ATP consumption. Furthermore, the ATP level in the ER is readily depleted by oxidative phosphorylation (**OxPhos**) inhibitors and that ER protein misfolding increases ATP uptake from mitochondria into the ER. These findings suggest that ATP usage in the ER may increase mitochondrial OxPhos while decreasing glycolysis, i.e. an '*anti-Warburg*' effect.

DOI: https://doi.org/10.7554/eLife.49682.001

**\*For correspondence:** rkaufman@sbpdiscovery.org

**Competing interests:** The authors declare that no competing interests exist.

## Introduction

Energy supply is a fundamental requirement for all cells to perform their biochemical functions. Universally, ATP is the single most important energy-supplying molecule in every form of life. ATP regeneration from ADP takes place in mitochondria mainly through OxPhos, and in the cytosol through glycolysis. Despite its heavy demand for ATP to facilitate protein folding and trafficking, the ER is not known to possess an independent ATP regeneration machinery. A membrane protein, '*ER-ANT*', is involved in the ATP translocation across the ER membrane, of which the biochemical properties are analogous to the mitochondrial **A**denosine **N**ucleotide **T**ransporter (**ANT**) (*Clairmont et al., 1992*; *Shin et al., 2000*). Other than that, little is known about how ATP gets into the ER in a living cell or whether/how ATP consumption is regulated in the ER lumen. For example, only one report described the genetic identification of an ATP transporter *ER-ANT1* in *Arabidopsis* which is restricted to plants and its deletion caused a disastrous plant phenotype, characterized by drastic growth retardation and impaired root and seed development (*Leroch et al., 2008*). The mammalian ER ATP transporter remained elusive until a recent publication identified SLC35B1/AXER as the putative mammalian ER ATP transporter (*Klein et al., 2018*).

ER ATP is essential to support protein chaperone functions for protein folding, such as BiP/GRP78, and trafficking (*Dorner et al., 1990*; *Braakman et al., 1992*; *Dorner and Kaufman, 1994*; *Wei et al., 1995*; *Rosser et al., 2004*). In fact, the level of ER ATP determines which proteins are able to transit to the cell surface (*Dorner et al., 1990*; *Dorner and Kaufman, 1994*). Although the

level of ER ATP is suggested to impact protein secretion, this has not been demonstrated, nor have the factors that regulate ATP levels in the ER been clearly elucidated, although an association with ER $Ca^{2+}$ pool was suspected (*Vishnu et al., 2014*; *Klein et al., 2018*). More recently, organelle specific ATP status determination was made possible with the genetically encoded FRET-based ATP reporter proteins targeted to select intracellular organelles, namely the mitochondrial localized *AT1.03* and the ER localized *ERAT4* probes (*Imamura et al., 2009*; *Vishnu et al., 2014*). A recent study revealed that the regulation of mitochondrial matrix ATP is highly dynamic and complex (*Depaoli et al., 2018*). Here, we studied ATP dynamics within the ER organelle in intact cells. Specifically, we monitored real-time changes in ATP levels inside the ER lumen in response to well-characterized OxPhos and/or glycolysis inhibitors in living Chinese hamster ovary (**CHO**), rat insulinoma INS1 and human Hela cells, at the single cell level using an ERAT-based FRET assay. In addition, we monitored the change in ER ATP upon $Ca^{2+}$ release from the ER, and further evaluated the ER ATP status in response to varying cytosolic $Ca^{2+}$ concentrations. From our findings we propose that cytosolic $Ca^{2+}$ attenuates mitochondrial-driven ATP transport into the ER lumen through a *CaATiER* (**Ca$^{2+}$-A**ntagonized **T**ransport **i**nto **ER**) mechanism. This model was further validated by knocking-down *Slc35b1* in HeLa, CHO and INS1 cells, and under conditions of ER protein misfolding in CHO cells.

## Results

### ER ATP comes from Mitochondrial OxPhos in CHO cells

Traditional ATP analytical methods based on biochemical or enzymatic assays inevitably require ATP liberation from endogenous compartments, and do not reflect compartment-specific ATP dynamics. Nevertheless, there is ample evidence supporting that differential ATP levels exist in membrane-bound organelles that use independent regulatory mechanisms in a compartment-specific manner (*Akerboom et al., 1978*; *Depaoli et al., 2018*; *Imamura et al., 2009*; *Vishnu et al., 2014*). To detect ATP levels in the ER lumen in vivo, (note that we use in vivo to indicate in a live cell) we expressed an ER-localized ATP sensor ERAT (ERAT4.01$^{N7Q}$) in H9 CHO cells engineered to induce mRNA expression of human clotting factor VIII (**F8**), encoding a protein which misfolds in the ER lumen, upon increased transcription promoted by histone deacetylase inhibition (*Dorner et al., 1989*; *Malhotra et al., 2008*). Confocal analysis of ERAT fluorescence (*Figure 1A*, green) revealed nearly complete co-localization with the ER marker, ER-Tracker Red (*Figure 1A*, red), as well as with the endogenous ER-resident protein PDIA6 detected by immunofluorescence (*Figure 1—figure supplement 1A*). Induction of F8 by SAHA treatment, an HDAC inhibitor, for 20 hr did not change the ER localization of the ERAT reporter (*Figure 1B*, and *Figure 1—figure supplement 1B*). Another protein ER marker, ER-RFP, also shows nearly complete co-localization with ERAT fluorescence although ER-RFP formed aggregates (*Merzlyak et al., 2007*) upon SAHA induction (*Figure 1—figure supplement 2A and B*, with RFP aggregates indicated with yellow arrow heads). As there is no known ATP regeneration machinery in the ER, and intracellular ATP regeneration from ADP takes place in mitochondria through OxPhos, and in the cytosol through glycolysis, it is of key importance to determine whether the ER ATP status depends on OxPhos and/or glycolysis (illustrated in *Figure 1C*). For this purpose, we analyzed ER ATP levels after OxPhos inhibition or glycolysis inhibition by monitoring the FRET signal through flow-cytometry, since this technique provides robust population statistics and circumvents photo-bleaching of the fluorescent probes. In order to achieve consistent ATP signal detection from a homogenous population of cells, we further constructed a clone of H9 CHO (*H9-D2 cell*) with stably transfected ERAT4.01$^{N7Q}$ reporter (*Figure 1—figure supplement 3A and B versus* C and D). In addition, as inhibition of glycolysis may have adverse effects on OxPhos, we tested whether 2-deoxyglucose (**2-DG**) administration reduces mitochondrial respiration by measuring the cellular oxygen consumption rate (**OCR**) in H9 CHO cells using a *Seahorse XF24* respirometry platform. Importantly, with glucose and pyruvate supplemented in the medium to support mitochondrial respiration, 2-DG administration did not negatively affect the OCR, under both coupled and uncoupled respiration conditions (*Figure 1D* and *Figure 1—figure supplement 4A*), suggesting that glycolysis inhibition by 2-DG does not acutely impair OxPhos in CHO cells. Interestingly, oligomycin (**Oligo**) treatment immediately increased the extracellular acidification rate (**ECAR**) in H9 CHO cells, reflecting glycolysis upregulation to compensate for ATP shortage due to

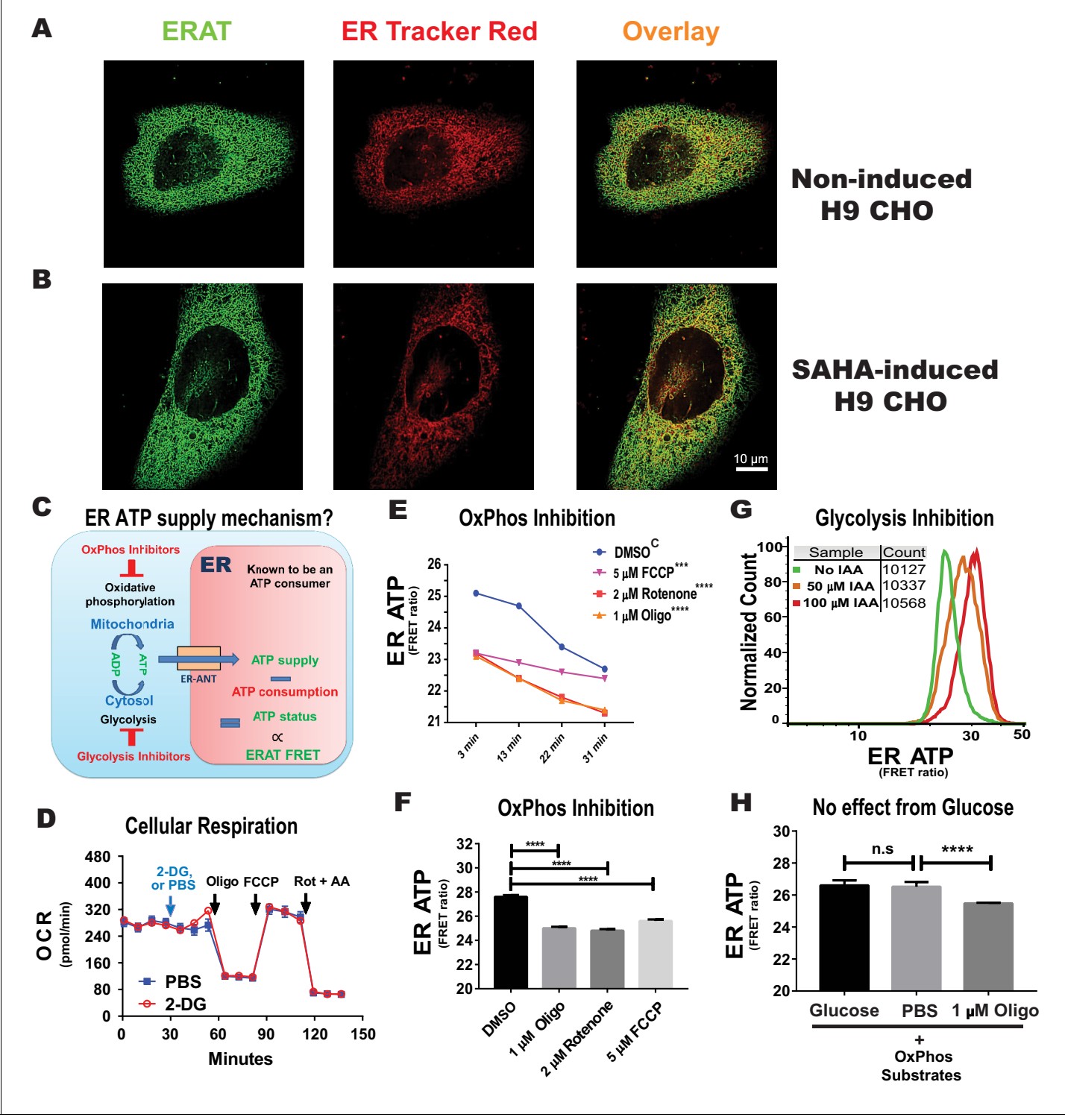

**Figure 1.** ER ATP homeostasis is maintained by oxidative phosphorylation. (**A**) Confocal microscopy confirms ER localization of the ER ATP FRET reporter, ERAT in H9-D2 CHO cells. A representative confocal micrograph shows a high degree of co-localization of ERAT fluorescence in green with ER-Tracker Red, in red. (**B**) Increased F8 expression in H9-D2 CHO cells by SAHA treatment does not alter ERAT's co-localization with ER-Tracker Red. (Scale bar: 10 μm) (**C**) A cartoon model depicting the ET ATP monitoring system. Importantly, the ER is an obligate ATP consumer as there is no known ATP regeneration system inside the ER lumen. ERAT reports steady state ATP levels inside the ER, which is a dynamic balance between ATP supply and ATP consumption. (**D**) 2-DG (20 mM) has no effect on cellular respiration. Oxygen consumption rates (OCR) in H9 CHO cells were measured by an *XF-24* platform (Seahorse BioScience), using serial injections of the compounds: first PBS or 2-DG followed sequentially by 1 μM oligomycin (Oligo), 1 μM

*Figure 1 continued on next page*

*Figure 1 continued*

FCCP and 1 µM rotenone together with 10 µM antimycin A (Rot + AA). (E) The effect of OxPhos inhibitors (oligomycin, rotenone and FCCP at the indicated concentrations) on ER ATP was examined by flow cytometry at four time points as indicated. The same tubes of suspended H9 CHO cells were repeatedly sampled after OxPhos inhibition. Statistical significance values for curve comparisons by two-way ANOVA are labeled for individual inhibitors versus the 'DMSO' control group marked by an upper case 'C'. (F) Administration of OxPhos inhibitors for 15 min, represented by oligomycin, rotenone and FCCP, rapidly reduces ER ATP in H9 CHO cells. DMSO was included as solvent control for comparison. Statistical significance values are labeled for individual comparisons. (G) Administration of IAA, a glycolysis inhibitor, increases ER ATP levels (detected by flow cytometry) in H9 CHO cells. Experiments were repeated four times, and a representative result is shown. Specifically, histograms of ER ATP levels for individual cells at two IAA concentrations (50 µM and 100 µM, treated for 2 hr) are shown with the y-axis representing normalized cell counts. More than ten thousand cells were sampled for every treatment condition, as indicated on the graph. (H) Glucose supplementation (5 mM) to DMEM medium with only OxPhos substrates (with 1 mM sodium pyruvate supplemented) does not affect ER ATP levels. H9 CHO cells stably transfected with the ERAT reporter were incubated in the serum-free modified DMEM medium, which contains substrates (i.e. pyruvate plus L-Ala-Gln) to support only OxPhos, for 1 hr at 37°C in a cell culture incubator, before flow analysis for ER ATP status. For the '1 µM Oligo' group, oligomycin was added to cells for the last 20 min of incubation in glucose-free medium. *Two-way ANOVA* was applied for statistical analysis of geometric means of fluorescence intensity (gMFI), with significance levels expressed as: *n.s*: - not significant; * - $p \leq 0.05$; **- $p \leq 0.01$; ***- $p < 0.001$; **** - $p < 0.0001$. The same statistical analysis was applied in the following figures.

DOI: https://doi.org/10.7554/eLife.49682.002

The following figure supplements are available for figure 1:

**Figure supplement 1.** The ERAT probe is localized to the ER lumen.
DOI: https://doi.org/10.7554/eLife.49682.003

**Figure supplement 2.** The ERAT probe is localized to the ER lumen.
DOI: https://doi.org/10.7554/eLife.49682.004

**Figure supplement 3.** A single clone of H9-D2 cells was engineered to facilitate flow-cytometry based ER ATP analysis.
DOI: https://doi.org/10.7554/eLife.49682.005

**Figure supplement 4.** 2-DG treatment does not reduce OxPhos of H9 CHO cells.
DOI: https://doi.org/10.7554/eLife.49682.006

**Figure supplement 5.** ER localization of ERAT probe is not affected by brief treatment with bio-energetic inhibitors.
DOI: https://doi.org/10.7554/eLife.49682.007

**Figure supplement 6.** The dynamic range of the ER ATP change was determined by flow cytometry.
DOI: https://doi.org/10.7554/eLife.49682.008

**Figure supplement 7.** Glucose supplementation does not alter ER ATP levels and 3-Bromopyruvate increases ER ATP levels.
DOI: https://doi.org/10.7554/eLife.49682.009

**Figure supplement 8.** Glucose supplementation increases total cellular ATP while 2-DG decreases the cytosolic ATP-to-ADP ratio.
DOI: https://doi.org/10.7554/eLife.49682.010

OxPhos inhibition, while 2-DG treatment inhibited this compensation (*Figure 1—figure supplement 4B*).

We subsequently tested three potent OxPhos inhibitors, oligomycin (targeting ATP synthase, also known as complex V), rotenone (targeting complex I), and FCCP (a protonophore that depolarizes the mitochondrial inner membrane to uncouple ATP synthase). All OxPhos inhibitors significantly decreased ER ATP levels immediately after administration, by analysis of ER ATP kinetics with time (*Figure 1E*) or by analysis at a single time point (15 min post treatment, *Figure 1F*). Importantly, the bio-energetic inhibitors used had no effect on ER localization of the ERAT probe, as shown by confocal microscopy in H9 CHO cells co-transfected with a cytosolic RFP-based reporter, Cyto TagRFP (*Merzlyak et al., 2007*) (*Figure 1—figure supplement 5A–F*), and was further reflected by no change in the *Pearson* correlation coefficient under these conditions (*Figure 1—figure supplement 5G*). As predicted by the chemical law of mass action, the decrease in ER ATP supply upon OxPhos inhibition should reduce ER ATP consumption. Furthermore, if we assume that the three OxPhos inhibitors do not stimulate ATP usage in the ER, the results (*Figure 1E and F*) indicate that the steady-state ER ATP supply is heavily dependent on OxPhos. Since the FRET signal generated from the ERAT reporter is ratiometric and does not provide a direct calibration corresponding to ATP concentration in the ER in live cells, we attempted to determine the effect of complete inhibition of cellular ATP regeneration thereby providing the lowest detection range of the ERAT reporter by treating cells for prolonged time (2 hr) with 2-DG, followed by oligomycin treatment (*Figure 1—figure supplement 6A and B*). This condition successfully identified the lower limit of the dynamic range for the ERAT reporter when the ratiometric FRET/CFP signal no longer decreases.

In contrast to OxPhos inhibition, the interpretation of effects from glycolysis inhibition is rather complicated, as there is no pure 'glycolysis inhibitor' per se, for example 2-DG application consumes cytosolic ATP and may even increase intracellular ADP levels as a substrate to boost mitochondrial ATP production through OxPhos (*Figure 1D*, and *Figure 1—figure supplement 4A*). With this caveat in mind, it was particularly intriguing to observe that incubation with iodoacetamide (**IAA**), an inhibitor of glyceraldehyde phosphate dehydrogenase (**GAPDH**), for 2 hr increased steady state ER ATP levels in H9 CHO cells, in a dose-dependent manner (illustrated in *Figure 1G*). In an effort to further measure the extent by which glycolysis-derived ATP contributes to the ER ATP pool, we found that glucose supplementation had no effect on ER ATP levels, in a defined medium with substrates supporting only OxPhos (*Figure 1H*), while oligomycin strongly decreased ER ATP under these conditions (p<0.0001, with representative traces of raw FRET ratios shown in *Figure 1—figure supplement 7A*). In contrast, a third widely used glycolysis inhibitor, 3-bromopyruvate (**3-BrP**), produced a similar ATP increase in the ER, in a dose dependent manner (*Figure 1—figure supplement 7B*). At the same time, the kinetics of the ER ATP increase by 3-BrP was distinct from that by IAA, suggesting compound-specific off-target effects by the 'glycolysis inhibitors' (*Figure 1—figure supplement 7C*). Finally, the observations made in H9 CHO cells were completely reproduced in CHO cells that do not express human F8 (*Urlaub and Chasin, 1980*), that is oligomycin or rotenone administration reduced steady state ER ATP levels (*Figure 1—figure supplement 7D*, indicated by red and orange curves). In contrast, inhibitors of glycolysis, 2-DG or IAA, immediately increased ER ATP levels (*Figure 1—figure supplement 7D*, indicated by dark green and light green curves).

To further determine whether and to what extend the intracellular ATP regeneration requires glycolysis, total cellular ATP content was further measured by luciferase assays under the same medium conditions. As expected, glucose supplementation significantly increased cellular ATP measured (*Figure 1—figure supplement 8A*), and 2-DG decreased cellular ATP, while oligomycin only had a minor effect on cellular ATP compared to 2-DG (*Figure 1—figure supplement 8B*). These observations are consistent with the notion that CHO cells are highly dependent on glycolysis to regenerate ATP, and that the OxPhos inhibition can be readily rescued by glycolysis upregulation (*Figure 1—figure supplement 4B*). Furthermore, 2-DG immediately decreased the cytosolic ATP/ADP ratio measured by flow-cytometry using the ratiometric *PercevalHR* probe (*Figure 1—figure supplement 8C*), while OxPhos inhibition by oligomycin did not significantly alter this ratio (*Figure 1—figure supplement 8D*), as previously shown by others (*Imamura et al., 2009*; *Tantama et al., 2013*). These results provide strong evidence supporting the notion that distinct ATP supplying and regulatory mechanisms exist for subcellular compartments within a cell, as we reported (*Depaoli et al., 2018*). During the flow cytometry data analysis, we noticed a reproducible decrease in steady-state ER ATP levels in the control group of H9-D2 CHO cells, especially after 30 mins of cell suspension (examples as shown in *Figure 1E*). We suspect this baseline ATP drift could be caused by the flow cytometry procedure, as extracellular matrix-attachment is crucial for optimal mitochondrial respiration (*Grassian et al., 2011*; *Schafer et al., 2009*). We acknowledge this feature as a disadvantage for flow-based ER ATP assay, and therefore have also included single-cell fluorescent microscopy recordings to complement our findings.

To conclude, OxPhos-derived ATP constitutes the major ATP supply source for the ER compartment, while it is not possible to conclude whether ER ATP depends on glycolysis-generated ATP.

## ER ATP import requires a Ca$^{2+}$ gradient across the ER membrane

Protein misfolding in the ER is experimentally induced with sarco/endoplasmic reticulum Ca$^{2+}$-ATPase (**SERCA**) inhibitors such as thapsigargin (**Tg**), which may disrupt ER ATP homeostasis. Consistent with this notion, application of the reversible SERCA inhibitor, 2,5-di-(tert-butyl)−1,4-hydroquinone (**BHQ**), caused a rapid decrease in ER ATP in rat insulinoma INS1 cells (*Figure 2A*). INS1 cells were selected because they secrete large amounts of another protein, proinsulin and insulin. As we reported previously (*Vishnu et al., 2014*), due to reduced ATP levels in the ER lumen, the FRET fluorescence intensity from the ERAT probe decreased while the corresponding CFP fluorescence intensity increased simultaneously (*Figure 2B*). The average of FRET over CFP ratios in the population of INS1 cells decreased and reached a plateau after 10 min post BHQ perfusion (*Figure 2C*), and simultaneous measurement of ER Ca$^{2+}$ by the D1ER probe (*Palmer et al., 2004*) in another set of INS1 cells confirmed the specific effect of BHQ on decreasing the ER Ca$^{2+}$ concentration (*Figure 2D*). We observed that ER ATP levels did not completely recover after BHQ removal, while in the same time

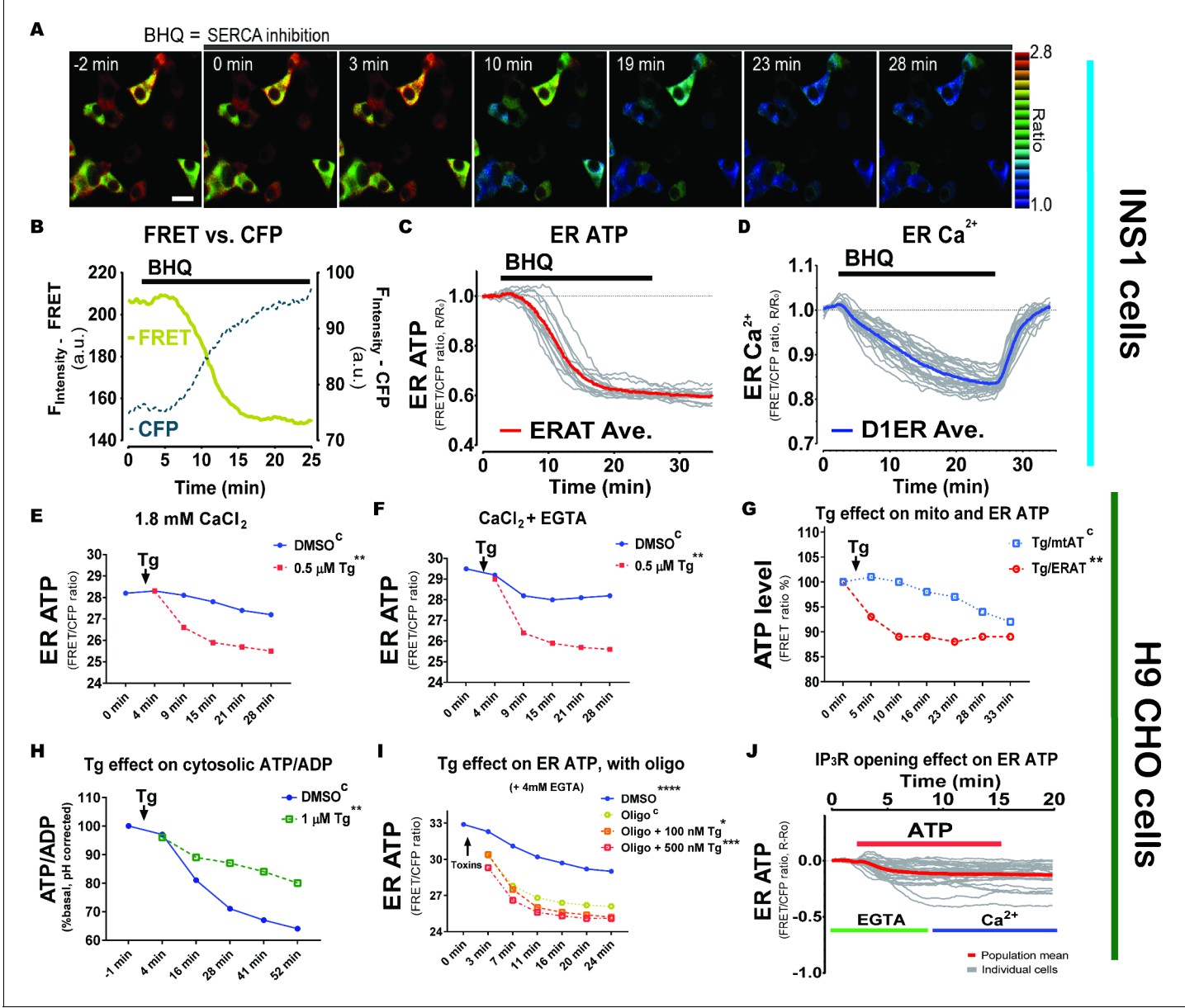

**Figure 2.** A $Ca^{2+}$ gradient across the ER membrane is required to maintain ER ATP homeostasis. (**A**) SERCA inhibition reduces ER ATP levels in adherent INS1 cells. BHQ (15 µM) was added at 0 min as indicated. Scale bar represents 10 µm. (**B**) A representative trace of the FRET signal overlaid with the CFP fluorescence intensity is shown for illustration purposes in an adherent INS1 cell. (**C**) BHQ reduces ER ATP. Y-axis represents the ratio between FRET intensity versus CFP fluorescence intensity. The average ratio is shown in red bold solid line in the graph, with individual INS1 cells shown in gray (n = 13 cells from three experiments). (**D**) Quantification of the effect of BHQ on ER $Ca^{2+}$ levels in individual INS1 cells. The average ratio from the $Ca^{2+}$ reporter (D1ER) is shown by a red solid line, with individual cells shown in gray (n = 25 cells from three experiments). Notably, BHQ withdrawal at 25 min by perfusion causes $Ca^{2+}$ refilling in the ER by SERCA re-activation. (**E**) Thapsigargin (Tg, 0.5 µM added at 0 min) reduces ER ATP significantly. Flow analysis was performed on H9 CHO cells containing the ERAT reporter and with 1.82 mM $CaCl_2$ in the culture medium. Statistical significance is labeled in the figure compared to the '*DMSO*' group as control, indicated with an upper case 'C'. (**F**) In the same experiment as shown in panel (**E**), when extra-cellular $Ca^{2+}$ in the medium was chelated by adding 4 mM EGTA, Tg shows the same effects as shown in panel (**E**) that is Tg decreases ER ATP significantly. (**G**) In response to Tg (0.25 µM final concentration, added at 0 min), the ER ATP concentration in H9 CHO decreases significantly faster than the decrease in mitochondrial ATP. ER ATP levels were measured by the ERAT probe and mitochondrial ATP was measured by the *mtAT 1.03* probe. ATP levels in the two compartments were measured simultaneously in two sets of H9 CHO cells and shared the same readout as the FRET versus CFP ratio of the probes ($F_{530}/F_{445}$). The ATP levels measured by FRET ratios were further used to calculate the Tg-induced ATP decrease standardized to the respective DMSO-treated control groups, and ATP levels are expressed as percentage of the '*DMSO*' group (% of DMSO). (**H**) Tg treatment (1 µM final concentration) increases the cytosolic ATP to ADP ratio in H9 CHO cells monitored by the *PercevalHR* reporter. Raw $F_{488}/F_{405}$ values from *PercevalHR* were corrected for the cytosolic cpYFP signal ($F_{488}/F_{405}$) in a different set of H9 CHO cells to compensate for the

*Figure 2 continued on next page*

*Figure 2 continued*

fluorescence change from pH change in the cytosol in response to Tg or DMSO. (I). In the presence of oligomycin, Tg-induced ER $Ca^{2+}$ depletion decreases ER ATP more quickly compared to oligomycin (1 µM) alone, and the effect is dependent on the Tg concentration, that is 500 nM Tg induced a faster decrease than 100 nM Tg. (J) Addition of extracellular ATP (100 µM) decreases the ER ATP concentration in intact H9 CHO cells, in the presence (green line) or absence (blue line) of $Ca^{2+}$ chelation by EGTA. In all panels, the ER ATP concentration was measured by the *ERAT4* (N7Q) probe and the y-axis indicates the FRET ratio derived by dividing fluorescence intensity at 530 nm ($F_{530}$) by that at 445 nm ($F_{445}$), with a 405 nm laser excitation, that is FRET ratio = $F_{530}/F_{445}$. Statistical significance is shown for the indicated concentrations, compared to the control group indicated by the superscript "C".

DOI: https://doi.org/10.7554/eLife.49682.011

The following figure supplements are available for figure 2:

**Figure supplement 1.** ER ATP levels in INS1 cells decrease after BHQ-mediated SERCA inhibition and recovery takes time.

DOI: https://doi.org/10.7554/eLife.49682.012

**Figure supplement 2.** Tg and CPA decrease ER $Ca^{2+}$ and ATP in H9 CHO cells due to SERCA inhibition.

DOI: https://doi.org/10.7554/eLife.49682.013

frame, the ER $Ca^{2+}$ concentration returned to its basal level (compare *Figure 2C–2D*). It is particularly interesting that the ER ATP levels took longer to recover when we monitored the INS1 cells for up to 4 hr after BHQ withdrawal (*Figure 2—figure supplement 1A and B*). Furthermore, the recovered ER ATP levels in the same INS1 cells were reduced by a second BHQ stimulation, in an extent proportional to the levels of ER ATP recovery (compare *Figure 2—figure supplement 1B and C*). These phenomena suggest that the $Ca^{2+}$-responsiveness of the putative ER ATP transporter has a 'memory' that lasts longer than the ER $Ca^{2+}$ perturbation. In a similar manner, SERCA inhibition by Tg, an irreversible SERCA inhibitor, rapidly decreased ER ATP levels in H9 CHO cells, reflected by a decrease in the ratio of FRET-to-CFP fluorescence intensity measured by flow cytometry (*Figure 2E*). SERCA inhibition by Tg decreased ER ATP, independent of whether the extracellular medium contained $Ca^{2+}$ (alpha MEM medium, 1.8 mM) (*Figure 2E*) or no extracellular $Ca^{2+}$ (by 4 mM EGTA chelation, *Figure 2F*). A third SERCA inhibitor, cyclopiazonic acid (**CPA**), induced a similar decrease in ER ATP at a dosage fifty times higher than Tg, presumably due to its low potency for inhibiting SERCA (*Figure 2—figure supplement 2A*). The decrease in ER ATP by Tg correlated well with ER $Ca^{2+}$ depletion as monitored by the GEM-CEPIA1er probe (*Suzuki et al., 2014*) (*Figure 2—figure supplement 2B*). Mechanistically, Tg disrupts the ER to cytosol $Ca^{2+}$ gradient by inhibiting SERCA, a $Ca^{2+}$ pump that shuttles 2 moles of $Ca^{2+}$ from the cytosol into the ER lumen at the expense of 1 mole of ATP (*MacLennan et al., 1997*). Since maintaining low cytosolic $Ca^{2+}$ concentrations (~100 nM under physiological conditions) also requires the action of plasma membrane $Ca^{2+}$ pumps (**PMCA**) that consume ATP, it is possible that the decrease in ER ATP upon Tg treatment reflects mitochondrial ATP depletion due to excessive ATP usage in the cytosol. This possibility was tested by simultaneously measuring ATP levels in two sets of H9 CHO cells expressing either the ERAT probe (*Figure 2—figure supplement 2C*) or the mitochondrial ATP probe *mtAT* (*Imamura et al., 2009*) (*Figure 2—figure supplement 2D*). Comparison of the decrease in relative ATP levels triggered by Tg revealed that the decrease in ER ATP preceded the decrease in mitochondrial ATP (*Figure 2G*), refuting the notion that Tg decreases ER ATP by decreasing mitochondrial ATP. In contrast, when the cytosolic ATP/ADP status was monitored by a structurally unrelated reporter, *PercevalHR* (*Tantama et al., 2013*), Tg treatment attenuated the decrease in cytosolic ATP/ADP ratio in H9 CHO cells (*Figure 2H*). Reduced cytosolic ATP usage by SERCA or reduced ER ATP supply (see '*CaATiER model*' in *Figure 3H*) may explain the effect of Tg on cytosolic ATP/ADP. Tg-mediated depletion of the ER $Ca^{2+}$ store did not appear to increase ATP usage in the ER since oligomycin treatment together with Tg reduced ER ATP only modestly, albeit significantly, compared to oligomycin alone (*Figure 2I*). In addition, we investigated if $Ca^{2+}$ release mediated through $IP_3R$ channel opening can affect ER ATP status in single cells, using a perfusion system and FRET-based fluorescence microscopy. Extracellular ATP is a well-known physiological $IP_3$-generating agonist through its engagement with purinergic receptors ubiquitously expressed on CHO cells. Indeed, exogenously supplemented ATP in the perfusion buffer noticeably decreased the ER ATP pool (*Figure 2J*). Furthermore, the effect of exogenous ATP was neither reversed by addition of extracellular $Ca^{2+}$ (indicated by the blue bar in *Figure 2J*) or washout of the agonist, again demonstrating that $Ca^{2+}$ mobilization lowers ER ATP levels in a sustained manner and supporting a 'memory' effect

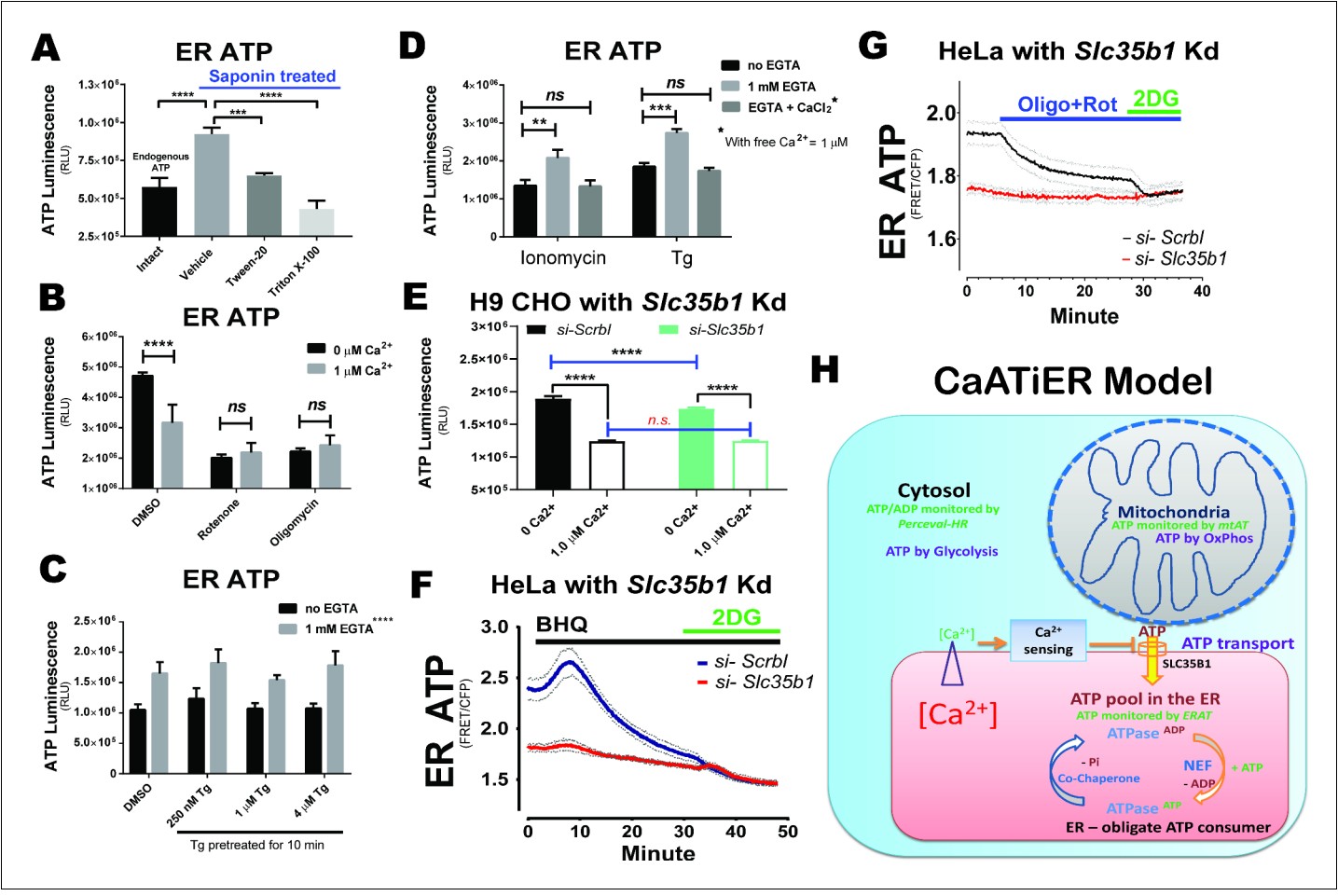

**Figure 3.** ATP transport from mitochondria to ER is inhibited by cytosolic $Ca^{2+}$. ATP stores in plasma membrane permeabilized H9 CHO cells were measured by a reporter-free method. H9 CHO cells were permeabilized with 75 μg/mL saponin in a bathing solution (referred to as 'respiration solution' hereafter). ATP production through OxPhos was supported by added pyruvate and malate (10 mM and 1 mM, respectively) as TCA substrates together with exogenous ADP (2 mM). The system was validated in two ways: First, the ATP store in the ER generated in permeabilized CHO cells was higher than that for the endogenous ATP pool detected in intact H9 CHOs (p<0.0001 between the 'Intact' and 'Vehicle' group). In addition, the ER ATP store was released within 5 min after treatment with Tween-20 (0.02% vol/vol, p=0.0007 compared to 'Vehicle' group) or Triton X-100 (0.02% vol/vol, p<0.0001), both after 25 min of respiration (**A**). Second, inhibition of OxPhos by oligomycin (1 μM) or rotenone (1 μM) reduces the ER ATP store to baseline levels. In addition, $Ca^{2+}$ supplemented (in the form of $CaCl_2$) at 1 μM final significantly reduced the ER ATP store (**B**). (**C**). SERCA ATPase activity is blocked by pre-incubating intact H9 CHO cells with increasing concentrations of Tg (250 nM, 1 μM, and 4 μM) for 10 min. Subsequently, cells were permeabilized as described above to test if $Ca^{2+}$ depletion affects ER ATP generated through OxPhos in permeabilized CHO cells. EGTA supplemented at 1 mM significantly increased the ER ATP store (p<0.0001). No significant difference was observed for Tg pre-treated groups. (**D**). When ionomycin (10 μM) or Tg (1 μM) were added at the same time with PM permeabilization, EGTA significantly increased ER ATP while further $Ca^{2+}$ add-back in the form of $CaCl_2$ abolished the EGTA effect on increasing the ER ATP store. (**E**). The $Ca^{2+}$ responsiveness of ER ATP levels was attenuated by siRNA-mediated *Slc35b1* knock-down in H9 CHO cells. Using the semi-intact cell system, CHO cells with *Slc35b1* knock-down by siRNA (24hrs post transfection) had significantly reduced ER ATP when no $Ca^{2+}$ was present in the buffer (****: p<0.0001). With 1 μM $Ca^{2+}$ in the buffer, ER ATP showed no difference between the control and the Slc35b1 knock-down group. (**F**). The $Ca^{2+}$ responsiveness of ER ATP levels by SERCA inhibition was attenuated by siRNA-mediated *Slc35b1* knock-down in *HeLa* cells. *HeLa* cells expressing the ERAT4 reporter were treated with siRNA against *Slc35b1* for 48 hr before cell imaging, with si-*Slc35b1*/siRNA control = 9.5 ± 1.4% quantified by qRT-PCR (mean ± S.E.M., n = 3 samples/group). Basal ER ATP levels significantly decreased in the si-*Slc35b1* group (p<0.001 by two-tailed t-test), compared to scrambled siRNA treated HeLa cells. Meanwhile the ER ATP decrease after BHQ addition was attenuated in the si-*Slc35b1* group. Glucose (10 mM) replacement by 2-DG (10 mM) in the end ensured cellular ATP depletion as another control. No difference was found for the two groups after 2-DG addition. (**G**) The ER ATP decrease after OxPhos inhibition (by oligomycin and rotenone) was abrogated by siRNA-mediated *Slc35b1* knock-down in HeLa cells. No difference was found for the two groups after 2-DG addition. For panels F and G, results are shown as traces representing mean ± S.E.M. (**H**) The cartoon depicts the *CaATiER* model showing mitochondrial ATP transport into the ER is antagonized by $Ca^{2+}$ on the cytosolic side of the ER membrane.

DOI: https://doi.org/10.7554/eLife.49682.014

The following figure supplements are available for figure 3:

*Figure 3 continued on next page*

*Figure 3 continued*

**Figure supplement 1.** Permeabilized H9 CHO cells produce ATP through OxPhos and ATP is detected in the respiration buffer.
DOI: https://doi.org/10.7554/eLife.49682.015

**Figure supplement 2.** In complement to *Figure 3E*.
DOI: https://doi.org/10.7554/eLife.49682.016

**Figure supplement 3.** Similar to the result in *Figure 3F*,.
DOI: https://doi.org/10.7554/eLife.49682.017

**Figure supplement 4.** Slc35b1 transcript was quantified by qRT-PCR, and the knock-down efficiency was > 90% in HeLa cells (**A**), and ~ 33% in INS1 cells (**B**).
DOI: https://doi.org/10.7554/eLife.49682.018

**Figure supplement 5.** The *CaATiER* mechanism responds to cytosolic $Ca^{2+}$ in a physiological range.
DOI: https://doi.org/10.7554/eLife.49682.019

**Figure supplement 6.** Since the respiration buffer contains 5 mM $MgCl_2$ as a basal ingredient, an additional 5 to 10 mM $MgCl_2$ was added to final concentrations of '*10 mM*' and '*15 mM*' as indicated, to test the $Mg^{2+}$ effect on ER ATP.
DOI: https://doi.org/10.7554/eLife.49682.020

of the putative $Ca^{2+}$ responsive ER ATP transporter. To summarize, ATP import into the ER apparently requires a $Ca^{2+}$ gradient ($Ca^{2+}_{ER} > Ca^{2+}_{cyto}$) across the ER membrane maintained by SERCA activity.

## Cytosolic $Ca^{2+}$ inhibits ATP import into the ER

The observations presented in *Figure 2* indicate that a $Ca^{2+}$ gradient is required to facilitate ATP transport into the ER lumen, reminiscent of the ATP/ADP exchange process mediated by the mitochondrial adenosine nucleotide transporters that is facilitated by a proton gradient across the mitochondrial inner membrane. However, the ER membrane is known to be quite 'leaky' to ions, with the $Ca^{2+}$ gradient requiring maintenance by vigorous SERCA $Ca^{2+}$ pumping into the ER lumen. As a consequence, it is theoretically difficult for the ER membrane to build a significant chemical potential to drive ATP/ADP exchange. Alternatively, as $Ca^{2+}$ depletion from the ER lumen is inevitably coupled with a cytosolic $Ca^{2+}$ increase in an intact cell (*Bagur and Hajnóczky, 2017*), the ER ATP transporter could respond to and be regulated by cytosolic $Ca^{2+}$, in a process described as *CaATiER*.

To test whether ER ATP is sensitive to cytosolic $Ca^{2+}$, we employed an independent classical ER ATP measurement for adherent H9 cells, facilitated by a standard luminescence-based ATP assay (*Akerboom et al., 1978*; *Burgess et al., 1983*). The plasma membrane (**PM**) was first permeabilized with low concentrations of saponin, a cholesterol solubilizing detergent, leaving the organelle membranes intact (as these membranes contains very low levels of cholesterol) to analyze ATP transport between mitochondria and ER. ATP was generated de novo by supplying mitochondrial respiration substrates (pyruvate plus malate) together with 2 mM ADP in the respiration buffer. At the end of the incubation, the respiration buffer containing ATP generated by OxPhos was removed by aspiration (which accounted for > 75% of the total ATP, see below). Since the ATP pool trapped in the mitochondrial matrix cannot be eliminated technically, it is possible the mitochondrial pool of ATP contributes to the 'trapped' ATP pool. However, for the permeabilized CHO cell system, we found that $Ca^{2+}$ or Tg addition did not affect mitochondrial respiration (*Figure 3—figure supplement 1A to D*), ensuring equal ATP generation. The same results further indicate that mito-matrix ATP content does not change in response to $Ca^{2+}$ or Tg, as the ATP synthase is extremely sensitive to the ATP-to-ADP ratio in the mitochondrial matrix. This conclusion was further demonstrated by carboxyatractyloside (an inhibitor of ANT proteins, hereafter abbreviated as '**CAt**') mediated respiration inhibition (*Figure 3—figure supplement 1A* to **D,** post CAt addition). Theoretically, it is unlikely that OxPhos would respond to exogenous $Ca^{2+}$ added in our assays, since greater than 1 μM $Ca^{2+}$ is needed to open the MICU-regulated MCU gating of $Ca^{2+}$ entry (*Waldeck-Weiermair et al., 2015*). Furthermore, the exogenously added ADP (2 mM) should effectively drive OxPhos-generated ATP out of the mitochondrial matrix into the ER lumen or the respiration buffer. Lastly, we verified that $Ca^{2+}$ addition did not affect total ATP generated in the supernatant (*Figure 3—figure supplement 1E*) calibrated by an ATP standard curve shown alongside for the absolute amount of ATP generated in our permeabilized CHO cell system (*Figure 3—figure supplement 1F*).

Using this system, we confirmed that only ATP trapped in the ER compartment accounts for the changes of luminescence signals above background (i.e., *ER ATP*), which was readily liberated by brief treatment with lipid-dissolving detergents (*Figure 3A*, p<0.001 and<0.0001 for Tween-20 and Triton X-100, respectively). In addition, ATP generated by OxPhos from the permeabilized cells was evidenced by the higher *ER ATP* contents trapped inside PM-permeabilized cells, compared to the endogenous ATP found inside PM-intact cells (*Figure 3A*, under '*Intact*' group). We also verified that ATP produced in this system was through OxPhos, as addition of OxPhos inhibitors at the beginning of incubation reduced ATP luminescence to background levels (*Figures 3B* and *1* μM '*Rotenone*' or '*Oligomycin*'). Interestingly, addition of $Ca^{2+}$ (1 μM) significantly decreased *ER ATP* (*Figure 3B*, p<0.0001 for '0 μM $Ca^{2+}$' vs. '1 μM $Ca^{2+}$' bars in the '*DMSO*' group), suggesting that cytosolic $Ca^{2+}$ inhibits ATP import from mitochondria into the ER. Alternatively, an increase in SERCA ATPase activity by increased cytosolic $Ca^{2+}$ may reduce ATP available for the ER ATP trans-porter. To rule out this possibility, Tg was applied to intact H9 cells briefly before PM-permeabiliza-tion, at increasing doses from 0.25 μM to 4 μM, and the $Ca^{2+}$ chelator EGTA was added to the respiration buffer to remove $Ca^{2+}$ leaked from the ER into the cytosol. While 1 mM EGTA signifi-cantly increased the *ER ATP* content, addition of Tg had no effect (*Figure 3C*). We also tested another $Ca^{2+}$ releasing chemical, ionomycin – a mobile $Ca^{2+}$ ionophore that permits $Ca^{2+}$ transfer down the gradient, with 1 μM Tg added as a control group without pretreatment of intact cells. Again, the only effect we detected was with $Ca^{2+}$ manipulation on the 'cytosolic' side of the ER membrane, that is EGTA chelation increased ER ATP and adding $CaCl_2$ back decreased the ER ATP (*Figure 3D*, with free $Ca^{2+}$ calculated to be 1 μM in presence of 1 mM EGTA).

A recent report (*Klein et al., 2018*) identified SLC35B1/AXER as the putative mammalian ER ATP transporter, which enabled us to validate our model using an *Slc35b1* knock-down strategy. Consis-tent with the proposed role of an ER ATP/ADP translocase, knock-down of *Slc35b1* in CHO H9 cells significantly decreased the baseline ER ATP level only in assay buffer containing no $Ca^{2+}$, but not in presence with 1 μM $Ca^{2+}$ (*Figure 3E*, p<0.0001 for '0 $Ca^{2+}$' and non-significant for '1.0 μM $Ca^{2+}$', respectively). The specificity of the ER ATP change was further confirmed by measuring ATP parti-tioned into the supernatant of the semi-intact CHO H9 system (*Figure 3—figure supplement 2A*), and by comparison of ER ATP as percentage of total ATP produced (*Figure 3—figure supplement 2B*) where it was estimated the total ATP produced is about 0.2–0.5 mM using the ATP standard (*Figure 3—figure supplement 2C*). For CHO H9 cells, *Slc35b1* knock-down efficiency was > 50%, in the presence or absence of Tm (*Figure 3—figure supplement 2D*). Similarly, using FRET-based microscopy in intact HeLa cells, we observed that the ER ATP decrease, after an initial increase, upon BHQ treatment was attenuated in *HeLa* cells upon *Slc35b1* knock-down (by siRNA to 9.5%, compared to control siRNA treated cells) (*Figure 3F*). In addition, we observed a transient increase in ER ATP shortly (0–10 min) after SERCA inhibition, probably due to ER ATP usage arrest as previ-ously reported (*Vishnu et al., 2014*). Similarly, *Slc35b1* knock-down also abolished the ER ATP decrease as a result of OxPhos inhibition (*Figure 3G*, by oligo plus rotenone). A similar significant baseline ER ATP decrease after *Slc35b1* knock-down was also observed for INS1 and H9 CHO cells (*Figure 3—figure supplement 3*) using microscopy, while *CaATiER* was proportionally attenuated. Furthermore, the varying *Slc35b1* knock-down efficiency may explain the difference in baseline ER ATP decrease for the three cell lines with HeLa > H9 CHO>INS1, as shown in (*Figure 3—figure sup-plements 2D* and *4*), suggesting SLC35B1 is a *bona fide* ATP/ADP translocase for the ER confirming Klein et al. Interestingly, using semi-intact H9 CHO cells, the *CaATiER* mechanism responded to $Ca^{2+}$ concentrations ranging from 500 nM to 2 μM (*Figure 3—figure supplement 5A and B*), with ATP content in the supernatant showing the opposite changes (*Figure 3—figure supplement 5C*, calibration provided in *Figure 3—figure supplement 5D*), further demonstrating that $Ca^{2+}$ concen-trations tested in our assay affected only ATP partitioning between ER and cytosol, but not mito-chondria ATP production (*Figure 3—figure supplement 1A*). Finally, the ER ATP decrease with increased $Mg^{2+}$concentration was negligible within a physiological range from '*5 mM*' to '*10 mM*' (p=0.21 by *post-hoc* t-test), confirming the decrease in ER ATP was specific to the $Ca^{2+}$ cation. The effect of adding $CaCl_2$ back was not due to increased $Cl^-$ concentration, as $Cl^-$ in the form of $MgCl_2$ in great excess over $CaCl_2$ only had a minor effect (*Figure 3—figure supplement 6*).

In summary, based on the above results, we propose that $Ca^{2+}$ inhibits ER ATP uptake (*Figure 3H*). In the *CaATiER* model, ATP from mitochondria is preferentially transported into the ER, possibly through a family of ER ATP transporter proteins represented by SLC35B1 (*Klein et al.,*

2018), and the cytosolic $Ca^{2+}$ concentration within this micro-domain attenuates ATP transport through a $Ca^{2+}$ sensing component. In addition, based on the *CaATiER* sensitivity to low μM of $Ca^{2+}$ (*Figure 3—figure supplement 5A and B*), we predict that SLC35B1 does not encode the $Ca^{2+}$-responsiveness considering the lack of $Ca^{2+}$-responsiveness from 10 to 50 μM when SLC35B1 was ectopically expressed in *E Coli* (*Klein et al., 2018*).

## Protein misfolding in the ER increases both ER ATP dependence on OxPhos and Tg-triggered $Ca^{2+}$ mobilization into mitochondria

Protein misfolding in the ER is a cellular stress condition characterized by activation of the unfolded protein response (**UPR**) (*Schröder and Kaufman, 2005*). Although ER stress is typically induced by pharmacological means, for example with Tg to deplete ER $Ca^{2+}$ store, or with tunicamycin (**Tm**) to inhibit N-linked glycosylation, we applied an ER stress induction model by expressing a misfolded protein in the ER lumen. ER stress was successfully induced by treating H9 CHO cells with HDAC inhibitors, such as SAHA or sodium butyrate (**NaB**), to increase F8 transcription from an HDAC-sensitive artificial promoter (*Dorner et al., 1989*) (*Figure 4A*). To better understand this proteostasis stress, we investigated if ATP supply and $Ca^{2+}$ homeostasis are affected upon induction of F8. As ER stress is often associated with $Ca^{2+}$ leak from the ER (in addition to those deliberately induced with Tg), we first validated this observation by investigating ER-initiated mitochondrial $Ca^{2+}$ influx, by employing the mitochondrial matrix-localized $Ca^{2+}$ reporter, mtGEM-GECO1 (*Zhao et al., 2011*) (*Figure 4B*). Pretreatment with 2-aminoethoxydiphenyl borate (**APB**) successfully blocked Tg-triggered mitochondrial $Ca^{2+}$ influx, although we cannot rule out the ability of APB to block other $Ca^{2+}$ channels (*Bootman et al., 2002*). As expected, the reporter promptly detected ionomycin-induced $Ca^{2+}$ entry into the mitochondrial matrix (*Figure 4C*). Although ER stress induced by SAHA treatment did not alter mitochondrial matrix $Ca^{2+}$ levels under resting conditions, Tg-triggered mitochondrial $Ca^{2+}$ influx was significantly more prominent in ER stressed H9 cells (*Figure 4C and D*). From these results, we propose that ER protein misfolding stimulates ER to mitochondria $Ca^{2+}$ transfer. The increased $Ca^{2+}$ transfer into mitochondria was not due to an increased ER $Ca^{2+}$ store since neither the O-cresolphthalein (**OCPC**) chromogenic method nor the ER $Ca^{2+}$ reporter (*GEM-CEPIA1er*) revealed a significant increase in the ER $Ca^{2+}$ content in SAHA-induced H9 CHO cells (*Figure 4E and F*), suggesting that the increased $Ca^{2+}$ entry into the mitochondrial matrix under ER stress conditions is either due to enhanced $Ca^{2+}$ mobilization from extracellular $Ca^{2+}$ (*Kaufman and Malhotra, 2014*; *Rizzuto et al., 2004*) or the close proximity between the ER and mitochondria under prolonged ER stress (*Bravo et al., 2011*). At the same time, oxygen consumption (by OCR) was not significantly different between ER stressed H9 CHO cells induced to express F8 and their un-induced counterparts (*Figure 4G*). Finally, when F8 expression was induced by SAHA treatment, ER ATP homeostasis became more dependent on OxPhos, as oligomycin treatment caused greater reduction in ER ATP for SAHA-treated H9 CHO cells compared to the un-induced H9 CHO cells (*Figure 4H*). This observation was not due to altered mitochondrial ATP supply (*Figure 4G*) as mitochondrial ATP levels were not affected upon induction of F8 expression (*Figure 4—figure supplement 1A and B*). Lastly, we verified the oligomycin-induced decrease in ER ATP in adherent single H9 CHO cells using FRET-based fluorescence microscopy (*Figure 4I& J*). Consistent with the ER ATP status measured by flow cytometry, the ER ATP decrease in response to complete OxPhos inhibition was significantly more pronounced in SAHA-induced ER stressed H9 CHO cells than in vehicle-treated CHO cells (quantified by maximal ER ATP reduction, *Figure 4K*, p<0.05). Since ATP dependence on OxPhos suggested that ER stress could have caused an 'anti-Warburg effect' by inhibiting ATP production from glycolysis, we further tested if the cytosolic bioenergetic dependence on OxPhos was similarly affected by ER stress induced by Tm. Similar to the ER compartment, the cytosolic ATP/ADP ratio decreased more quickly upon OxPhos inhibition (*Figure 4—figure supplement 1C and D*), while no obvious difference was detected for the mitochondrial matrix ATP pool. Fully supporting the 'anti-Warburg effect', ER stressed H9 CHO cells had significantly reduced glucose uptake independently of the F8 inducer, measured by 2-NBDG fluorescence assay (*Figure 4—figure supplement 2A and B*). Notably, only H9 CHO but not 10A1 CHO showed a significant reduction in glucose uptake in response to SAHA, as the latter cells lack the HDAC inhibitor responsiveness and do not exhibit ER stress, as previously demonstrated (*Dorner et al., 1989*). To summarize, ER stress induced by a misfolded ER luminal protein causes metabolic changes, characterized by increased $Ca^{2+}$ exchange between ER and mitochondria (*Bagur and Hajnóczky, 2017*; *Kaufman and*

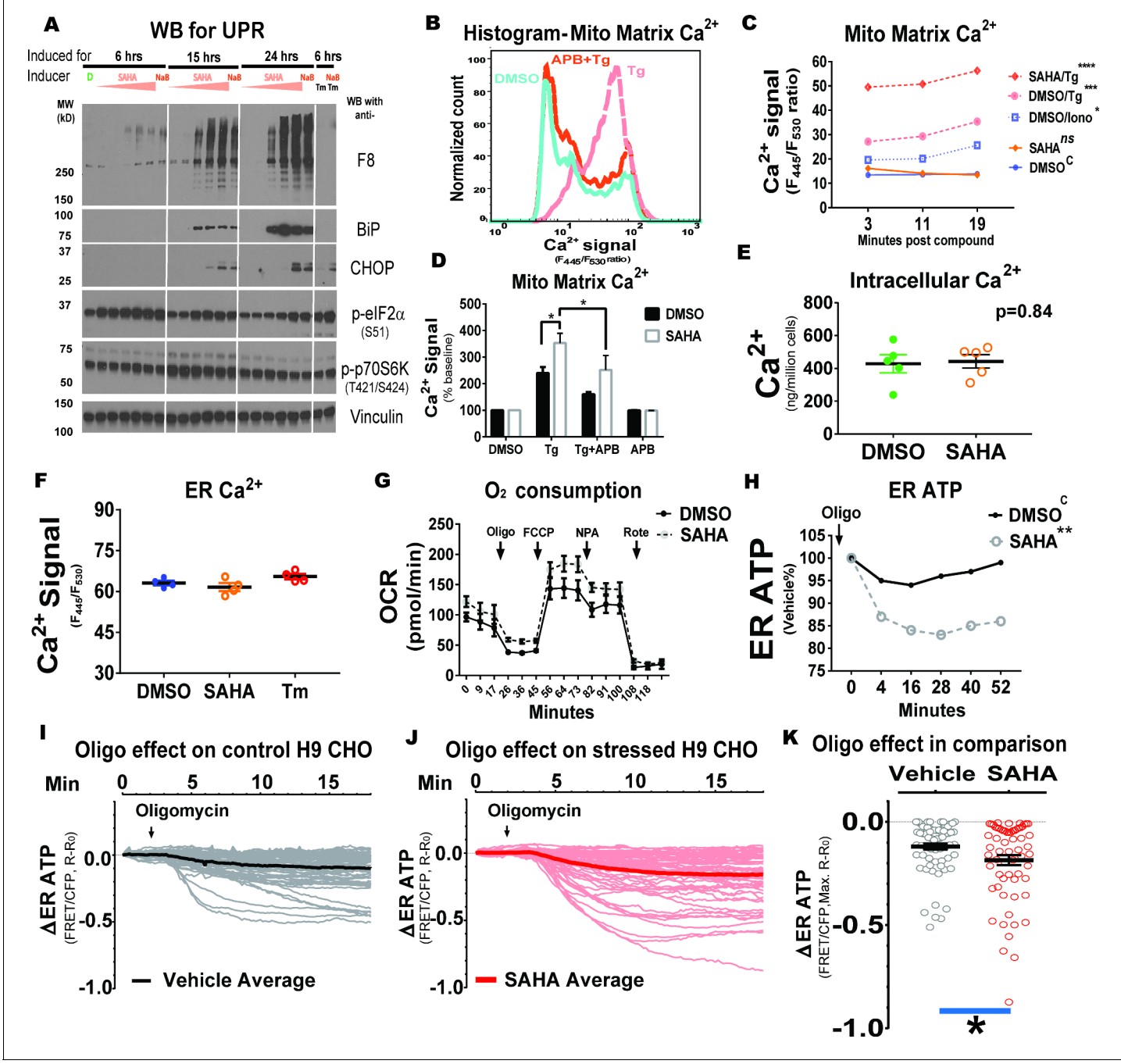

**Figure 4.** Protein misfolding in the ER increases ER ATP dependence on OxPhos and increases ER to mitochondrial $Ca^{2+}$ trafficking. (A) When treated with HDAC inhibitors, such as SAHA or sodium butyrate (NaB), F8 transcription is induced in H9 CHO cells accompanied by UPR activation. SAHA was tested at increasing concentrations, at 50 nM, 200 nM, 1 µM, 5 µM and 20 µM, as indicated by pink arrow heads. NaB was used at 5 mM. 'D' indicates the same volume of DMSO as a vehicle control. The last two lanes received 10 µg/mL tunicamycin (Tm) as positive controls for ER stress. Treatment times are labeled on the top of the Western blot panels. (B). Representative histograms of mitochondrial $Ca^{2+}$ influx, reported by the *mtGEM-GECO1* $Ca^{2+}$ probe, were overlaid for visual comparison. For the particular 'Tg' trace, H9 CHO cells were treated with 5 µM Tg for 30 min; and for the 'APB+Tg' trace, the same cells received 50 µM of APB before Tg stimulation. (C) While the mitochondrial $Ca^{2+}$ concentrations show no difference at the basal level, Tg (5 µM)-triggered a higher mitochondrial $Ca^{2+}$ spike in H9 cells with induced F8 expression (5 µM SAHA x 21 hr). An independent group of un-induced H9 CHO cells received ionomycin (10 µM) as a positive control. (D) Tg (5 µM)-triggered mitochondrial $Ca^{2+}$ spikes in un-induced H9 and induced H9 cells for F8 expression are shown by bar graphs, with average signal strengths represented as percent of baseline (without Tg treatment) from four consecutive readings within 45 min. The $Ca^{2+}$ signal from the mitochondrial matrix in ER stressed H9 cells (5 µM SAHA x 21 hr) was significantly increased compared to its vehicle-treated control group ('SAHA' versus 'DMSO', p<0.05), both of which were partially blocked by 40 µM

*Figure 4 continued*

APB. (**E**) Intracellular Ca$^{2+}$ content was not significantly changed in H9 cells with F8 expression induced by SAHA, compared to control H9 cells as measured by the OCPC chromogenic method. (**F**) ER Ca$^{2+}$ levels in H9 CHO cells were monitored by the *GEM-CEPIA1er* probe through flow cytometry-based ratiometric analysis. SAHA (5 μM for 18 hr) induces ER stress by increasing F8 expression in the ER, and Tm (100 ng/mL for 18 hr) induces ER stress by blocking N-linked glycosylation on proteins translocated into the ER. ER Ca$^{2+}$ levels in H9 CHO cells remained unchanged compared to the DMSO-treated control cells. (**G**) In H9 CHO cells with induced F8 expression, the OCR was slightly increased but not significantly different, as measured by the Seahorse flux assay (n = 6 wells/group). (**H**). Blockade of ATP production through OxPhos by oligomycin depletes ATP more quickly in induced H9 cells compared to un-induced H9 CHO cells, when standardized to DMSO-treated cells. (**I**) and (**J**) Traces for ER ATP in individual H9 CHO cells are shown in response to oligomycin administration (10 μM, indicated by arrow heads). In panel (**I**), cells were treated 18 hr with DMSO, a vehicle control for SAHA; In panel (**J**), cells were treated 18 hr with 5 μM SAHA. (**K**) Similar to measurements made by flow-cytometry (panel **H**), oligomycin caused significantly greater maximal ER ATP reduction (*, p<0.05 by two-tailed *Student's* t-test) in SAHA-induced H9 cells compared to '*Vehicle*' treated H9 CHO cells.

DOI: https://doi.org/10.7554/eLife.49682.021

The following figure supplements are available for figure 4:

**Figure supplement 1.** While mitochondrial matrix ATP levels remain unchanged in ER stressed H9 cells, upon OxPhos inhibition the cytosolic ATP/ADP ratios decrease more quickly than unstressed H9 cells.

DOI: https://doi.org/10.7554/eLife.49682.022

**Figure supplement 2.** Glucose uptake in ER stressed H9 CHO cells was reduced compared to non-stressed H9 CHO cells.

DOI: https://doi.org/10.7554/eLife.49682.023

*Malhotra, 2014*), increased dependence on ATP produced by OxPhos and by decreased cellular glucose uptake.

## Discussion

ATP drives most metabolic reactions inside the cell, and its transport mechanism between mitochondria and cytosol is well characterized. In contrast, little is known about how ATP is transported into the ER, an obligate ATP consuming organelle with no known ATP regeneration machinery (*Braakman et al., 1992*; *Dorner et al., 1990*).

### Mitochondria supply ATP to the ER and cellular OxPhos is enhanced by ER ATP usage

By using a series of ATP and ATP/ADP ratio reporters to monitor real-time ATP changes in sub-cellular compartments and by complementary direct measurements of ER ATP, we observed that ATP in the ER is mainly supplied from mitochondria and that ATP homeostasis in the ER is maintained through a cytosolic Ca$^{2+}$-antagonized ATP transport mechanism on the ER membrane (*Figure 3H*). It was surprising that the ATP level in the ER is insensitive to transient glucose deprivation and inhibition of glycolysis, which causes an immediate reduction in cytosolic ATP. This result is consistent with our conclusion that OxPhos supplies ATP to the ER and a highly efficient ATP transport mechanism exists from mitochondria to ER. Considering the close proximity between the ER and mitochondria (*Scorrano et al., 2019*), it is possible that ATP exiting mitochondria immediately traffics through the ER ATP transporter(s), whereas ATP produced by glycolysis is spatially more distant to be efficiently captured by the ER. In addition, supporting the symbiotic evolutionary hypothesis of mitochondria, our findings suggest that the ER takes special advantage of ATP-producing mitochondria to fulfill the energetic demands for protein folding, trafficking and secretion (*Scorrano et al., 2019*), reflected by the exquisite sensitivity of the ER ATP levels to OxPhos inhibition. Therefore, an intimate cross-talk between ER protein misfolding and mitochondrial function could have evolved as a bioenergetic advantage for eukaryotes to develop into multi-cellular organisms, in which cell-to-cell communications is necessitated by paracrine and endocrine signals. Based on our findings, we propose that ATP usage in the ER is a key signaling event that controls mitochondrial bioenergetics, in addition to the signaling mediated by Ca$^{2+}$ and reactive oxygen species (**ROS**) (*Kaufman and Malhotra, 2014*).

## Cytosolic Ca$^{2+}$ attenuates ATP import into the ER

One of our most important observations is that a Ca$^{2+}$ gradient across the ER membrane is necessary for ATP transport into the ER. Specifically, to demonstrate that cytosolic Ca$^{2+}$ inhibits ATP import, we applied an artificial PM-permeabilized cell system, by forcing mitochondrial respiration to supply ATP, for the reason that 1) the Ca$^{2+}$ concentration can be conveniently manipulated in this semi-intact experimental system; 2) the ATP supply is no longer limited by ADP availability which is considered a rate-limiting factor in vivo. Our *CaATiER* model was further validated by knock-down of ER ATP transporter expression, encoded by the *Slc35b1* gene (*Klein et al., 2018*). Notably, the SERCA inhibitor, BHQ, induced a phenomenal ER ATP decrease, which was greatly attenuated in HeLa cells with *Slc35b1* knock-down (*Figure 3F*), indicating that the observed ER ATP drop is mediated by reduced ATP import into the ER. The results confirm that 1) ATP is not freely permeable to the ER membrane, and 2) The ATP transport mechanism on the ER membrane (through SLC35B1/ AXER) is exquisitely sensitive to the cytosolic Ca$^{2+}$ concentration. Although it was suggested that the immediate Ca$^{2+}$ mediated increase in ER ATP requires AMPK activity (*Vishnu et al., 2014*), whether AMPK is involved in *CaATiER* in intact cells is an open question. Our findings lead us to propose that cytosolic Ca$^{2+}$ inhibits ATP import into the ER, indicating that a Ca$^{2+}$ responsive element faces the cytosolic surface (*Figure 3H*), analogous to the mitochondrial Ca$^{2+}$ uniporter (**MCU**) gating system (*Liu et al., 2016*; *Waldeck-Weiermair et al., 2015*).

Physiological processes require exquisitely meticulous and tight regulation of ATP, that is cytosolic ATP abundance is sensed and converted into a Ca$^{2+}$ chemical gradient by the coupling function of SERCA where excess cytosolic ATP will only be delivered to the ER when cytosolic Ca$^{2+}$ is at physiological levels (<500 nM). In summary, we propose that increased cytosolic Ca$^{2+}$ reduces ER ATP usage to conserve ATP for cytosolic functions. Evolutionarily, the risk of ER stress resulting from a transient ATP shortage is accommodated by bountiful ER chaperones, such as GRP78/BiP and GRP94, for which peptide binding activities are regulated by ATP (*Dorner et al., 1990*; *Rosser et al., 2004*; *Wei et al., 1995*), and Ca$^{2+}$ responsive chaperones (*Bagur and Hajnóczky, 2017*), such as calnexin and calreticulin (*Krebs et al., 2015*). Previous observations associating ER Ca$^{2+}$ release with a transient ER ATP increase (*Vishnu et al., 2014*) (and repeated in *Figure 3F*) could reflect a temporary reduction in ER ATP consumption, rather than reflecting the luminal Ca$^{2+}$ sensitivity of the ER ATP import machinery.

## ER protein misfolding increases the ER bioenergetic requirement for ATP from OxPhos

It has long been postulated that Ca$^{2+}$ micro-domains formed by ER-mitochondrial contacts could potentially stimulate mitochondrial respiration to encourage efficient ATP production, especially under conditions of ER stress (*Bravo et al., 2011*; *Kaufman and Malhotra, 2014*). The structural proximity is maintained by the ER-mitochondria tethering molecules, such as Mitofusin 2 (*Naon et al., 2016*; *Schrepfer and Scorrano, 2016*), while the functional cooperation through adenosine nucleotide exchange between the ER and mitochondria has not been directly proven. Uniquely here, we demonstrate that the ER preferentially depends on ATP produced from OxPhos, which was further confirmed by our observation that ER protein misfolding in H9 CHO cells causes an '*anti-Warburg*' effect. Whether this observation can be extrapolated to interpret cancer cell metabolism warrants further investigation. While confirming the intimate functional cooperation of ER and mitochondria through exchange of adenosine nucleotides, our findings demonstrate that increased Ca$^{2+}$ on the cytosolic surface of the ER membrane inhibits ER ATP import and predict that the Ca$^{2+}$ exchange domain is not compatible with the ATP exchange domain between mitochondria and the ER, suggesting the existence of either A) Two categories of contact domains between the organelles, or B) A domain that operates in two distinct modes (Ca$^{2+}$ exchange mode versus ATP exchange mode), which are temporally separated.

ER stress caused by protein misfolding can cause cell death under pathological conditions. ER stress-induced cell death is associated with increased oxidative stress (*Malhotra and Kaufman, 2007*; *Malhotra et al., 2008*). It is plausible that AMPK activation as a consequence of Ca$^{2+}$ efflux from the ER (*Klein et al., 2018*) can lead to compensatory mitochondrial ATP production in an attempt to resolve ER stress by supplying more ATP to the ER. OxPhos up-regulation as a feedback mechanism to increase ER ATP consumption could contribute to mitochondrial ROS generation,

although altered mitochondrial function was not detected in our study (*Figure 4G* and *Figure 4—figure supplement 1A and B*). Conversely, dysfunctional mitochondria can elevate ROS production by the ER (*Murphy, 2013*). In our CHO cell model system, when Tg was applied to release ER $Ca^{2+}$, the $Ca^{2+}$ influx into the mitochondrial matrix was significantly elevated in ER-stressed CHO cells, which could reflect the closer proximity between ER and mitochondria to ensure sufficient ATP uptake into the ER (*Kaufman and Malhotra, 2014*). In vivo, a similar process of IP$_3$R-mediated $Ca^{2+}$ release in response to IP$_3$ generated by repeated plasma membrane receptor engagement may exacerbate ROS production as a result of mitochondrial $Ca^{2+}$ overload (*La Rovere et al., 2016*).

To conclude, our results expand our knowledge of cellular bioenergetics by demonstrating that a highly efficient ATP supply mechanism exists between mitochondria and ER that is antagonized by cytosolic $Ca^{2+}$. Our findings should stimulate research to identify and to elucidate the $Ca^{2+}$-sensing molecular mechanism associated with the ER ATP transporter molecule(s).

# Materials and methods

## Key resources table

| Reagent type (species) or resource | Designation | Source or reference | Identifiers | Additional information |
|---|---|---|---|---|
| Cell line (*Cricetulus griseus*) | CHO DUK cell line | Dr. Lawrence Chasin at Columbia University, USA., Personal gift | N/A | Confirmed by karyotyping |
| Cell line (*Cricetulus griseus*) | CHO H9 cell line | Made in Kaufman Lab | N/A | Confirmed by karyotyping, further by F8 WB |
| Cell line (*Rattus norvegicus*) | INS1 cell line | Gift from Dr. Christopher B. Newgard, Duke University, USA | N/A | Confirmed by WB and electrophysiology |
| Cell line (*Homo-sapiens*) | HeLa S3 cell line | ATCC | # CCl-2.2 | Assurance by ATCC |
| Antibody | anti-F8, IgG1 (Mouse monoclonal) | Green Mountain Antibodies | GMA 012 | WB: 1:1000 |
| Antibody | anti-BiP, IgG (Rabbit monoclonal) | Cell Signaling Technology | CST 3177 | WB: 1:1000 |
| Antibody | anti-CHOP, IgG (Rabbit polyclonal) | Santa Cruz Biotechnology | SC 575 | WB: 1:1000 |
| Antibody | phospho-eIF2α (Ser51), IgG (Rabbit monoclonal) | Cell Signaling Technology | CST 3597S | WB: 1:1000 |
| Antibody | anti-phospho-p70S6K (Thr421/Ser424), IgG (Rabbit polyclonal) | Cell Signaling Technology | CST 9204 | WB: 1:1000 |
| Antibody | anti-VINCULIN, IgG1 (Mouse monoclonal) | Sigma | V9131 | WB: 1:1000 |
| Recombinant DNA reagent | *mtAT 1.03, plasmid* | Dr. Hiromi Imamura at Kyoto University, personal gift | N/A | ATP level |
| Recombinant DNA reagent | *ERAT 4.01 N7Q, plasmid* | NGFI, Austria | N/A | ATP level |
| Recombinant DNA reagent | pTagRFP-C, *plasmid* | *Evrogen* | #FP141 | Cytosolic location |
| Recombinant DNA reagent | GW1-*Perceval HR, plasmid* | Addgene | #49082 | ATP/ADP ratio |

*Continued on next page*

*Continued*

| Reagent type (species) or resource | Designation | Source or reference | Identifiers | Additional information |
|---|---|---|---|---|
| Recombinant DNA reagent | *D1ER, plasmid* | Dr. Demaurex at Universitè de Genève, Switzerland, personal gift | N/A | $Ca^{2+}$ level |
| Recombinant DNA reagent | pCIS *GEM-CEPIA1er, plasmid* | Addgene | #58217 | $Ca^{2+}$ level |
| Recombinant DNA reagent | CMV-mito-*mtGEM-GECO1, plasmid* | Addgene | #32461 | $Ca^{2+}$ level |
| Recombinant DNA reagent | *ER-RFP, plasmid* | Addgene | #62236 | ER localization |
| Recombinant DNA reagent | *cpYFP, plasmid* | Dr. Yi Yang at East China University of Science and Technology, personal gift | N/A | pH level |
| Sequence-based reagent | Puromycin resistant gene, 5' primer 5'-ACAAATGTGGTAAAATCGATAAGGATCCG-3'; | Integrated DNA Technologies | N/A | PCR primer |
| Sequence-based reagent | Puromycin resistant gene, 3' primer 5'-GAGCTGACTGGGTTGAAGGCT-CTCAAGGGC-3' | Integrated DNA Technologies | N/A | PCR primer |
| Sequence-based reagent | *siRNA Slc35b1-*5'-GAG ACU ACC UCC ACA UCA A dTdT-3' (targeting 3'-UTR of human gene) | Microsynth AG, Balgach, Switzerland | N/A | siRNA |
| Sequence-based reagent | *siRNA scrambled-*5'-AGG UAG UGU AAU CGC CUU G dTdT-3' (control for human *Slc35b1* knockdown) | Microsynth AG, Balgach, Switzerland | N/A | siRNA |
| Sequence-based reagent | *siRNA targeting mouse Slc35b1 sequence –*5'-CCACATGATGTTGAACATCAA-3' | Qiagen | Mm_Ugalt2_1, SI01461523 | siRNA |
| Sequence-based reagent | *siRNA targeting mouse Slc35b1 sequence –*5'- AAGAAGGTGGTTGGAATAGAA-3' | Qiagen | Mm_Ugalt2_2, SI01461530 | siRNA |
| Sequence-based reagent | *siRNA targeting mouse Slc35b1 sequence –*5'-TCGGTAAATCCTGCAAGCCAA-3' | Qiagen | Mm_Ugalt2_4, SI01461544 | siRNA |
| Sequence-based reagent | *siRNA scrambled – Sequence proprietary* | Qiagen | Allstars Negative Control siRNA, Cat #1027280 | siRNA |
| Commercial assay or kit | TransFast Transfection Reagent | Promega Corporation, Madison, USA | # E2431 | |
| Commercial assay or kit | FuGENE 6 Transfection Reagent | Promega | # E2693 | |
| Commercial assay or kit | Calcium Assay Kit | Adipogen Corp. | # JAI-CCA-030 | |
| Commercial assay or kit | XF24 extracellular flux assay kit | Seahorse BioSciences | # 100850–001 | |
| Commercial assay or kit | Ingenio Electroporation solution | Mirus Bio LLC | # MIR 20114 | |
| Commercial assay or kit | ATPLite kit | Perkin Elmer | # 6016943 | |

*Continued on next page*

*Continued*

| Reagent type (species) or resource | Designation | Source or reference | Identifiers | Additional information |
|---|---|---|---|---|
| Commercial assay or kit | MycoAlert plus Mycoplasma Detection kit | Lonza | #LT07-703 | |
| Chemical compound, drug | 2-Deoxy-D-glucose | Chem-Impex Int'l, INC | # 21916 | |
| Chemical compound, drug | 2,5-Di-(t-butyl)—1,4-hydroquinone (BHQ) | SIGMA | # 112976–25G | |
| Chemical compound, drug | 2-NBDG | Cayman Chemical | # 11046 | |
| Chemical compound, drug | 3-Bromopyruvate | Aldrich Chemistry | # 16490–10G | |
| Chemical compound, drug | Cyclopiazonic acid | Alfa Aesar | # J61594 | |
| Chemical compound, drug | FCCP | SIGMA | # C2920-10MG | |
| Chemical compound, drug | Iodoacetamide | SIGMA | # I1149-5G | |
| Chemical compound, drug | Ionomycin | AdipoGen | # AG-CN2-0418 | |
| Chemical compound, drug | Oligomycin A | Alfa Aesar | # J60211 | |
| Chemical compound, drug | Rotenone | SIGMA | # R-8875–1G | |
| Chemical compound, drug | SAHA | Tocris | # 4652 | |
| Chemical compound, drug | Thapsigargin | SIGMA | # T9033-5MG | |
| Chemical compound, drug | Tunicamycin | Santa Cruz Biotechnology | # sc-3506 | |
| Software, algorithm | GraphPad Prism 7 | GraphPad Software | Version: 7 | |
| Software, algorithm | FlowJo 10 | FlowJo LLC | Version: 10.2 | |
| Software, algorithm | Adobe Illustrator | Adobe | Version: CS 5.1 | |

## Construction and validation of H9 CHO cell line

The F8 inducible H9 CHO cell system was described previously (*Dorner et al., 1989*; *Malhotra et al., 2008*). Expression of F8 was confirmed by Western blot using specific monoclonal antibodies – anti-F8 (Green Mountain Antibodies, GMA 012, murine IgG1), anti-BiP (Cell Signaling Technology, CST 3177, rabbit IgG), anti-CHOP (Santa Cruz Biotechnology, SC 575, rabbit IgG), anti-phospho-eIF2$\alpha$ at Ser51 (Cell Signaling Technology, CST 3597S, rabbit IgG), anti-phospho-p70S6K at Thr421/Ser424 (Cell Signaling Technology, CST 9204, rabbit IgG), and anti-VINCULIN (Sigma, V 9131, murine IgG1).

## Construction of novel reporter cell lines

CHO cells were transfected with the reporter plasmids using the Amaxa Nucleofector II electroporation system in the Mirus Ingenio solution (Cat # MIR 20114). For reporter plasmids with G418 resistance, a brief G418 selection (0.5 mg/mL) was applied on transfected CHO cells for one week. The population of surviving cells was allowed to recover for two weeks before they were used for ER ATP analysis by flow cytometry. For reporter plasmids without a mammalian selection marker, a puromycin resistance cassette (~1.23 kb, amplified from plasmid pGL4.21) was co-transfected using the *Amaxa Nucleofector II* electroporation system in the *Mirus Ingenio* solution. After a brief

puromycin selection (5 µg/mL for five days), cells were allowed to recover for two weeks before they were used for fluorescence ratio analysis by flow cytometry.

More specifically, plasmids encoding the *ER-RFP* reporter (ER-mRFP, #62236), the *PercevalHR* reporter (GW1-PercevalHR, #49082), the *CEPIA1er* reporter (pCIS GEM-CEPIA1er, #58217) and the *mtGEM-GECO1* reporter (CMV-mito-GEM-GECO1, #32461) were purchased from *Addgene*. The plasmid encoding the *ERAT4.01^{N7Q}* reporter was purchased from *Next Generation Fluorescence Imaging* Company (NGFI, Austria). The plasmid encoding the *TagRFP* reporter was purchased from *Evrogen* (pTagRFP-C, #FP141). The plasmid encoding the *mtAT1.03* reporter was a kind gift from Dr. Hiromi Imamura at Kyoto University. The plasmid encoding the *cpYFP* reporter was a kind gift from Dr. Yi Yang at the East China University of Science and Technology, and the plasmid encoding the *D1ER* reporter was a kind gift from Dr. Demaurex at Universitè de Genève, Switzerland. Refer to *Supplementary file 1* for a summary of the reporter proteins' subcellular localization and their reporting specificity.

## Confocal microscopy

For live cell imaging of subcellular localization of ERAT in H9 CHO, targeting to the ER was validated using an array confocal laser scanning microscope (ACLSM) built on a fully automated inverse microscope (Axio Observer.Z1, Zeiss, Göttingen, Germany), using a 100x objective (Plan-Fluor 100 x/1.45 oil, Zeiss). CHO cells expressing ERAT were additionally transfected with ER-RFP (*Snapp et al., 2006*), and either treated with vehicle (DMSO) or 5 µM SAHA for 18 hr. Live cells were excited using diode lasers (Visitron Systems, Puchheim, Germany): CFP of ERAT was excited at 445 nm (50 mW), RFP of ER-RFP was excited at 561 nm (50 mW). Emissions were collected using the emission filters ET480/40 for CFP and E570LPv2 for RFP, respectively (Chroma Technologies Corporation, VT, USA). All images were captured at a binning of 2 using a photometrics CCD camera (CoolSnap HQ2, Photometrics, Arizona, USA), and images were processed using ImageJ software.

For fixed H9 CHO cells after immunostaining with anti-PDIA6 antibody, confocal images of stably transfected CHO cells were taken on an Olympus Fluoview 1000 platform, using the following excitation and emission filters - for DAPI, Ex 405 nm/Em 461 nm; for ERAT protein, Ex 488 nm/Em 519 nm; for PDIA6 (immuno-detected by Alexa-594 conjugated secondary antibody), Ex 561 nm/Em 618 nm. Objective lenses of 60x (PlanApo N, N.A = 1.42 oil) and 100x (UPlanSApo N.A. = 1.40 oil) were used.

## Quantification of ER ATP and Ca²⁺ levels by fluorescence ratiometry analysis in live cells

Analyses of ER ATP levels in INS-1 832/13 (INS-1) cells were facilitated with the ERAT 4.01 reporter introduced by conventional lipofectamine-transfection. Imaging of ER Ca²⁺ levels was performed using the ER targeted Ca²⁺ reporter D1ER. Prior to measurements, cells were equilibrated for 30 min using storage buffer composed of (mM): 138 NaCl, 5 KCl, 2 CaCl₂, 1 MgCl₂, 10 HEPES, 2.6 NaHCO₃, 0.44 KH₂PO₄, 0.34 Na₂HPO₄, 10 D-glucose, 2 L-glutamine, with the following supplements (vol/vol): 0.1% vitamins, 0.2% essential amino acids, and 1% penicillin–streptomycin, pH adjusted to 7.4 with NaOH. The experimental buffer used for the perfusion of the cells during the fluorescence microscopic experiments was composed of (mM) 138 NaCl, 5 KCl, 1 MgCl₂, 10 HEPES, 10 glucose, either with 2 CaCl₂ or 0.1 EGTA with no added CaCl₂, pH adjusted to 7.4 with NaOH. During the experiment, buffers were exchanged using a flow chamber, connected to a gravity-based perfusion system (NGFI, Graz, Austria) and a vacuum pump (Chemistry diaphragm pump ME 1 c, Vacuubrand, Wertheim, Germany).

CHO cells stably expressing the ERAT4.01^{N7Q} reporter were directly analyzed after removing the alpha MEM medium. Cells were cultured and analyzed in µ-slide four wells (ibidi GmbH, Planegg, Germany). For the experiments, alpha MEM was removed and replaced by experimental buffer composed of (mM): 138 NaCl, 5 KCl, 1 MgCl₂, 10 HEPES, 10 glucose, 2 CaCl₂, 2 pyruvate and four glutamate, pH adjusted to 7.4 using NaOH.

All fluorescence microscopic experiments were performed using an iMic inverted and advanced fluorescent Microscope with a 40x magnification objective (alpha Plan Fluor × 40, Zeiss, Göttingen, Germany) and a motorized sample stage (TILL Photonics, Graefling, Germany). Excitation was performed at 430 nm (Polychrome V, Till Photonics), emissions were collected simultaneously at 475 nm

for CFP and 525 nm for YFP, respectively, using a beam splitter. Images were captured using a binning of 2 with a CCD camera (Allied Vision Technologies, Stadtroda, Germany) and analysis was performed using Live Acquisition software (TILL Photonics). Representative ratio images were created using MetaMorph microscopy automation and image analysis software (Molecular Devices, Sunnyvale, CA, USA).

## ER and mitochondrial ATP level analysis by flow cytometry

For FRET-based ATP determination, a Novocyte 3000 flow cytometer (ACEA BioSciences) was used to record ERAT fluorescence at channels with Ex/Em filters set as following: 1). 405 nm/ 445 (band width: 45) nm; 2). 405 nm/ 530 (band width: 30) nm. ER and mitochondrial ATP levels for individual cells were defined as the ratio of fluorescence intensity of channel 2 (Ex/Em: 405/530) divided by that of channel 1 (Ex/Em: 405/445), similar to parameters used previously (*Imamura et al., 2009*; *Vishnu et al., 2014*). In addition, fluorescence intensities of the following channels were recorded for probe abundance quantification, and for data validity verification: 1) 405 nm/ 572 (band width: 28) nm; 2) 488 nm/ 530 (band width: 30) nm; and 3) 488 nm/ 572 (band width: 28) nm.

## Cytosolic ATP/ADP ratio analysis by flow cytometry

For cytosolic ATP/ADP ratio determination in H9 CHO cells expressing the *PercevalHR* reporter, an LSR Fortessa flow cytometer (BD Biosciences) or a Novocyte 3000 flow cytometer (ACEA BioSciences) were used to record fluorescence intensity of channels with Ex/Em as following: 1) 405 nm/ 525 nm (band width: 50 nm); and 2) 488 nm/ 510 nm (band width: 25 nm). The cytosolic ATP/ADP ratio for individual cells is defined as the ratio of fluorescence intensity of channel 2 (Ex/Em - 488/510) divided by that of channel 1 (Ex/Em - 405/525), similar to previously described (*Imamura et al., 2009*; *Tantama et al., 2013*). In addition, as fluorescence signals from the *PERCEVAL-HR* protein are known to be sensitive to cytosolic pH changes upon compound addition, the ratiometric $F_{525}/F_{510}$ was corrected by simultaneously measuring pH changes using the cpYFP overexpressing H9 CHO cells using a first-order correction (*Zhao et al., 2015*).

## Mitochondria $Ca^{2+}$ influx analysis by flow cytometry

CHO cells were transfected with the *mtGEM-GECO1* plasmid as described above, and after brief G418 selection for a week, cells were allowed to recover for two weeks before they were used for mitochondria $Ca^{2+}$ influx analysis by flow cytometry using a Novocyte 3000 flow cytometer. For mitochondrial matrix $Ca^{2+}$ level, a ratiometric measurement was derived using fluorescence intensity of channel one with Ex/Em of 405 nm/ 445 nm (band width: 45 nm) divided by that of channel two with Ex/Em of 405/530 nm (band width: 30 nm), similar to a procedure previously described (*Zhao et al., 2011*).

## ER luminal $Ca^{2+}$ analysis by flow cytometry

CHO cells were transfected with the *GEM-CEPIA1er* plasmid as described above, and after brief puromycin selection (5 μg/mL for five days), fifteen percent of the transfected cells were positive for the *CEPIA1er* reporter as quantified by flow cytometry analysis. Cells were allowed to recover for two weeks before they were used for ER lumen $Ca^{2+}$ analysis by flow cytometry facilitated by a Novocyte 3000 flow cytometer. For $Ca^{2+}$ determination, a ratiometric measurement was derived using fluorescence intensity of channel one with Ex/Em of 405 nm/ 445 nm (band width: 45 nm) divided by that of channel two with Ex/Em of 405/530 nm (band width: 30 nm), similar to a procedure previously described (*Suzuki et al., 2014*).

## Flow data analysis by FlowJo software

Flow cytometry data recorded by cytometers were saved as FCS 3.1 files and analyzed by FlowJO software (Version: 10.2, FlowJo LLC) on a Mac platform. Briefly, cells were first gated by forward light scatter and side light scatter (FSC-H and SSC-H, respectively) to ensure that only fluorescence signals from live singlet cells were analyzed. Within the population, subsets of reporter-positive and reporter-negative populations were further differentiated by plotting the fluorescence intensity at designated channels. A ratio parameter was further defined using the FlowJo's '*Derived Parameters*' function, as defined above for each reporter. Normality of the derived parameter was subsequently

checked by plotting a histogram for every ratiometric parameter. Furthermore, the geometric mean of fluorescence intensity (gMFI) was applied for population statistics analysis, as ratiometric parameters usually follow a log-normal distribution. Typically, statistics from a reporter positive population of more than 3000 singlet cells at any given time point were used to ensue data reproducibility.

## Effect of pharmacological reagents on ER and mitochondria ATP levels

Mito-toxins and glycolysis inhibitors were diluted to 100x stock concentrations in DMSO or PBS. All chemical inhibitors were purchased from Sigma or Thermo Fisher unless indicated otherwise. On the day of flow analysis, 10 µL of compounds were aliquoted into Falcon polypropylene tubes (Corning Cat # 352063) before 1 mL of cell suspension was added to make the final concentration. In any particular experiment where multiple compounds were used for compound effects comparison, for example *Figure 1E and F*, all compounds were pre-pipetted into the Falcon tubes before cell addition. To ensure population homogeneity, cells from the same tubes were aliquoted to the compound-containing tubes or to the vehicle-containing tubes for data collection.

Furthermore, based on the observation that H9 cells with high ERAT4 reporter expression have improved data sphericity for FRET ratio (*Figure 1—figure supplement 3A&B* versus **C and D**), we generated a single H9-D2 CHO cell clone with unanimous high levels of ERAT4 expression, by seeding single cells into multiple 96-well cell culture plates, followed by manual selection of clones with high YFP fluorescence intensities. Data sphericity for FRET ratio of H9-D2 CHO cell was further confirmed by flow cytometry-based FRET signal analysis, as described above.

## Cellular metabolic flux analysis by Seahorse XF24 platform

Cellular oxygen consumption rate (**OCR**) and extracellular acidification rate (**ECAR**) were analyzed on a Seahorse XF24 analyzer (*Seahorse BioSciences*). H9 CHO or CHO-DUK cells were seeded onto a XF24 cell culture microplate 4–5 hr before the assay to allow cell attachment to the plate. OCR and ECAR were measured following XF24's standard operating procedures as per manufacturer's manual, facilitated by a XF24 extracellular flux assay kit (*Seahorse BioSciences*, Part # 100850–001). In addition, to measure OCR and ECAR in F8-induced H9 CHO cells, the subject CHO cells were first induced by 5 µM SAHA for 21 hr in a 10 cm cell culture dish before re-plating into a XF24 cell culture microplate, to ensure equal cell input to the control un-induced H9 CHO cells.

For permeabilized CHO cell respirometry, cells were permeabilized with rPFO reagent (Agilent #102504–100) in a cytosol-like buffer containing 70 mM sucrose, 220 mM mannitol, 10 mM $KH_2PO_4$, 5 mM $MgCl_2$, 2 mM HEPES, 1 mM EGTA and 0.2% BSA(w/v), with pH adjusted to 7.4 by KOH.

The standardized procedure measuring mitochondrial respiration on an XF platform can be found at the following website: https://www.agilent.com/cs/library/technicaloverviews/public/5991-7157EN.pdf.

## ER ATP assay in saponin-permeabilized H9 CHO cells measured by luciferase assay

H9 CHO cells were seeded onto 96-well plates at a density of fifty thousand cells per well and were allowed to attach and grow overnight to reach eighty to ninety percent confluency. On the day of assay, cells were first permeabilized with artificial cytosol-like buffer containing 75 µg/mL saponin. The buffer contains 70 mM sucrose, 220 mM mannitol, 10 mM $KH_2PO_4$, 5 mM $MgCl_2$, 2 mM HEPES, 1 mM EGTA (unless when it was intentionally omitted) and 0.2% BSA(w/v), with pH adjusted to 7.4 by KOH. In addition, 10 mM pyruvate and 1 mM malate plus 2 mM ADP were added to maintain mitochondria respiration upon membrane permeabilization. $Ca^{2+}$ concentrations in cytosol-like buffer were controlled by adding increasing amounts of $CaCl_2$. Free $[Ca^{2+}]$ was estimated according to a protocol previously described (*Schoenmakers et al., 1992*) (*Supplementary file 2*). H9 cells were further incubated at 37°C in a cell culture incubator for 20–30 min, before the buffer was removed by taping the cell culture plate on a stack of dry paper towels. *ATPLite* kit (Perkin Elmer, Cat # 6016943) was used to quantify the ATP store in permeabilized H9 cells, following the manufacturer's protocol. Bioluminescence signals were recorded on a *SpectraMax i3x* multi-mode detection platform (Molecular Devices), with an integration time of 500 milliseconds per well. ATP measurements from eight to ten wells were averaged to plot the final graph under all conditions. Tg (0.25–1

µM) and oligomycin (1–2.5 µM) were used in inhibit SERCA and ATP synthase respectively, as indicated.

In addition, the same *ATPLite* kit was used to measure total cellular ATP. For *Figure 1H* and *Figure 1—figure supplement 7A* and *Figure 1—figure supplement 8A*, H9 CHO cells were incubated for 1 hr in serum-free DMEM medium containing substrates supporting only OxPhos (Seahorse Bio-Sciences, Cat. #100965 with 1 mM sodium pyruvate supplemented), in experiments where glucose supplementation effect was tested.

### *SLC35B1* knock-down in HeLa cells

For siRNA mediated knock-down of *SLC35B1* in HeLa cells, the following siRNA sequence, targeting the *SLC35B1* 3'-UTR, was used: 5'-GAG ACU ACC UCC ACA UCA A dTdT-3'. Control cells were treated with a scrambled siRNA of the following sequence: 5'-AGG UAG UGU AAU CGC CUU G dTdT-3'. Cells were transfected at a confluency of ~ 70% using TransFast transfection reagent (Promega GmbH, Mannheim, Germany) according to the following protocol: 1.5 µg ERAT4.01 plasmid, 0.12 nmol of siRNA against *SLC35B1* 3'-UTR or control siRNA and 3 µl of TransFast were added to 1 ml of FCS free DMEM medium. Mixture was incubated for 15 min at room temperature before applying onto adherent HeLa cells, replacing cell growth media containing 10% FCS. After 4 hr, transfection mixture was exchanged for DMEM plus 10% FCS. Cells were incubated for another 48 hr in a humidified incubator at 37°C prior to imaging analysis. Similar transfection procedures were used for INS1 and CHO cells using commercially available siRNAs from Qiagen.

### Detection of intracellular $Ca^{2+}$ levels by chromogenic assay in CHO lysates

Total intracellular $Ca^{2+}$ levels were measured by the O-Cresolphtalein (OCPC) chromogenic method, using components provided in a $Ca^{2+}$ Assay Kit (Adipogen Corp., Cat # JAI-CCA-030). Briefly, H9 CHO cells were grown to 70% confluency in 10 cm cell culture dishes and induced to express F8 with 5 µM SAHA for 22 hr. Cells were trypsinized, gently centrifuged and re-suspended in 1 mL complete culture medium. After cell density quantification, six million cells were dispensed into a separate set of 1.5 mL Eppendorf tubes for $Ca^{2+}$ quantification. Cells were pelleted by centrifugation, washed once with $Ca^{2+}$-free HBSS, and lysed immediately in 120 µL of 3% trichloroacetic acid on ice for 30 min, with intermittent vortexing. Cell lysates were further clarified by centrifugation at 6,000 rpm for 15 min using a bench-top Eppendorf Centrifuge (Model # 5415R). Cleared supernatants were used for OCPC-based $Ca^{2+}$ level detection as per manufacturer's instructions. Optical densities (OD at 570 nm) were read and recorded by a *VERSAmax* microplate reader (Molecular Devices) and $Ca^{2+}$ concentrations were determined by the $Ca^{2+}$ standard curve provided in the OCPC kit.

### Western blot analysis

CHO cells were grown on multiple-well plates and treated with compound in alpha MEM growth medium (*Dorner et al., 1989*). Cells were harvested by direct lysis with RIPA buffer (150 mM NaCl, 50 mM Tris pH 7.4, 1% Triton X100, and 0.5% Sodium Deoxycholate). Cell lysates were pre-cleared by centrifugation and protein concentration were determined by Bio-Rad's Dc protein assay kit. Total cellular protein (10–20 µg) was loaded onto a Bio-Rad's precast SDS-PAGE gel for electrophoresis and subsequent semi-dry transfer to a nitrocellulose membrane, as previously described[15]. Primary antibodies used for immunoblotting are summarized in '*Construction and validation of H9 CHO cell line*' section.

### Quantification and Statistical Analysis

Flow cytometry data recorded by cytometers were saved as FCS 3.1 files and analyzed by FlowJo software (Version: 10.2, FlowJo LLC) on a Mac platform. A ratio parameter was defined using the FlowJo's '*Derived Parameters*' function, as defined above for each reporter. Normality of the derived parameter was subsequently checked by plotting a histogram for every ratio parameter. Furthermore, the geometric means of fluorescence intensities (gMFI) were applied for population statistics analysis, as ratiometric parameters usually follow a log-normal distribution. Typically, statistics from a reporter positive population of more than 3000 single cells at any given time point were used to ensure data reproducibility.

Colocalization by *Pearson correlation* analysis of CHO cells expressing *ERAT4.01$^{N7Q}$* and *TagRFP* was performed using ImageJ software, with plugins '*Colocalization Test*' and '*Fay Randomization*' algorithms. Prior to the analysis, images were further 2D-deconvoluted using MetaMorph software (Molecular Devices, San Jose, USA).

Compartmental ATP and/or Ca$^{2+}$ levels were compared using the '*Ordinary Two-way ANOVA*' function provided by GraphPad's Prism software (Ver: 7.0), with alpha level of 0.05. Time and compounds were assumed two independent parameters for ANOVA analysis. *Dunnett's* multiple comparisons test was subsequently performed for comparison between individual groups versus diluent control (DMSO or PBS), or control group as indicated for particular experiments. Statistical significance was expressed as following: *n.s* - not significant; * - $p \leq 0.05$; **- $p \leq 0.01$; ***- $p < 0.001$; **** - $p < 0.0001$.

## Supplemental Information

Supplemental information includes pdf files for '*Supplementary file 1* and '*Supplementary file 2*'.

## Acknowledgements

We thank Drs. Markus Waldeck-Weiermair and Wolfgang F Graier from the Medical University of Graz (Austria) for sharing their critical comments and discussion of studies. We also thank Dr. Hiromi Imamura for sharing the *mtAT1.03* vector and Dr. Yi Yang for providing the *cpYFP* vector. Administrative support was kindly provided by Ms. Taryn Goode. Mrs. Yoav Altman (Flow Cytometry Core) and Leslie Boyd (Cell Imaging Core) at the SBP Medical Discovery Institute provided technical guidance and helpful discussion on imaging methodologies. Both cores receive support from the SBP NCI Cancer Center Support Grant P30 CA030199. This work was supported by the FWF Austrian Science Fund: P28529-B27 to RM and by NIH grants R01HL052173, R37DK042394, R24DK110973, R01DK103185, R01DK113171, R01AG062190 and R01CA198103 to RJK. RJK is a member of the UCSD DRC (P30 DK063491) and Adjunct Professor in the Department of Pharmacology, UCSD.

## Additional information

### Funding

| Funder | Grant reference number | Author |
|---|---|---|
| National Heart, Lung, and Blood Institute | R01HL052173 | Randal J Kaufman |
| National Institute of Diabetes and Digestive and Kidney Diseases | R37DK042394 | Randal J Kaufman |
| National Institute of Diabetes and Digestive and Kidney Diseases | R24DK110973 | Randal J Kaufman |
| National Institute of Diabetes and Digestive and Kidney Diseases | R01DK103185 | Randal J Kaufman |
| National Institute of Diabetes and Digestive and Kidney Diseases | R01DK113171 | Randal J Kaufman |
| National Institute on Aging | R01AG062190 | Randal J Kaufman |
| National Cancer Institute | R01CA198103 | Randal J Kaufman |
| National Cancer Institute | P30CA030199 | Randal J Kaufman |
| Austrian Science Fund | P28529-B27 | Roland Malli |
| National Institute of Diabetes and Digestive and Kidney Diseases | P30DK063491 | Randal J Kaufman |

The funders had no role in study design, data collection and interpretation, or the decision to submit the work for publication.

## Author contributions

Jing Yong, Conceptualization, Resources, Data curation, Formal analysis, Validation, Investigation, Methodology, Writing—original draft, Project administration, Writing—review and editing; Helmut Bischof, Data curation, Formal analysis, Validation, Investigation, Methodology, Writing—review and editing; Sandra Burgstaller, Formal analysis, Validation; Marina Siirin, Investigation; Anne Murphy, Conceptualization, Resources, Formal analysis, Methodology; Roland Malli, Conceptualization, Resources, Data curation, Supervision, Funding acquisition, Validation, Investigation, Methodology, Writing—original draft, Project administration, Writing—review and editing; Randal J Kaufman, Conceptualization, Resources, Formal analysis, Supervision, Funding acquisition, Investigation, Methodology, Writing—original draft, Project administration, Writing—review and editing

## Author ORCIDs

Jing Yong (iD) https://orcid.org/0000-0002-4970-408X
Helmut Bischof (iD) https://orcid.org/0000-0003-2380-600X
Roland Malli (iD) http://orcid.org/0000-0001-6327-8729
Randal J Kaufman (iD) https://orcid.org/0000-0003-4277-316X

## Decision letter and Author response

Decision letter https://doi.org/10.7554/eLife.49682.029
Author response https://doi.org/10.7554/eLife.49682.030

# Additional files

## Supplementary files

• Source data 1. Data sets for main figures.
DOI: https://doi.org/10.7554/eLife.49682.024

• Supplementary file 1. Summary table of reporters used in CHO cells and their intended specificity.
DOI: https://doi.org/10.7554/eLife.49682.025

• Supplementary file 2. Free $Ca^{2+}$ concentration estimates for $CaCl_2$ containing respiration buffers.
DOI: https://doi.org/10.7554/eLife.49682.026

• Transparent reporting form
DOI: https://doi.org/10.7554/eLife.49682.027

## Data availability

All data generated or analysed during this study are included in the manuscript and supporting files. Requests for reagents should be directed to and will be fulfilled by the Lead Contact, Randal J Kaufman (rkaufman@sbpdiscovery.org).

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
