## [Decision Letter]

Thank you for submitting your article "Mitochondria supply ATP to the ER through a mechanism antagonized by cytosolic Ca^2+^" for consideration by *eLife*. Your article has been reviewed by two peer reviewers, and the evaluation has been overseen by a Reviewing Editor and Suzanne Pfeffer as the Senior Editor. The following individuals involved in review of your submission have agreed to reveal their identity: Richard Zimmermann (Reviewer #2).

The reviewers have discussed the reviews with one another and the Reviewing Editor has drafted this decision to help you prepare a revised submission.

Summary:

The authors report new findings concerning the control of ER energy homeostasis by cytosolic calcium levels. They conclude that the ATP supply of the ER is mostly generated by mitochondrial OxPhos. In addition, they claim that cytosolic Ca^2+^ controls the ER ATP import, which is mediated by SLC35B1/AXER and that the elicitation of protein misfolding in the ER increases the ATP demand.

Essential revisions:

The major conclusions are of sufficient interest for publication in *eLife*, as the work addresses important aspects of ER biology. However, there are some issues that need to be addressed before the paper can be recommended for publication in *eLife*.

1) The link between *Slc35B1* and *CaATiER* is weak. The data on HeLa cells do not support the *CaATiER* model and there are no data on SLC35B1/AXER for the other two cell lines. Therefore, additional work is required to provide this link or, alternatively, to identify the ATP carrier that is linked to *CaATiER* in INS1 or CHO cells.

2) An in-depth analysis of the role of *Slc35B1* is needed to strengthen the study. It is required to show that *Slc35B1* is indeed the one and only ATP transporter to the ER. The analysis of a cytoplasmic calcium sensitivity/dependence of the *Slc35B1* may provide some useful information.

3) The discrepancies between the old and new results in INS1 cells need to be explained.

These criticisms of the work are listed in great detail in the reviews, which are attached in full. In addition, the reviewers offered many thoughtful and constructive suggestions to improve the manuscript.

Reviewer #1

In this manuscript, Kaufman, and colleagues reported the identification how ATP enters the ER lumen through a cytosolic calcium-antagonized mechanism, or, as proposed by the authors a *CaATiER* (Ca-Antagonized Transport into ER). They report that a calcium gradient across the ER membrane is necessary for ATP transport into the ER suggesting that cytosolic calcium signaling is coupled with ATP transport to the ER. They propose an attractive explanation that cytosolic increased calcium inhibits ATP import into the ER lumen. This is to reduce ATP consumption by ER. They also report that ATP abundance in the ER and traffic from mitochondria is sensitive to inhibitors of the oxidative phosphorylation and ER protein misfolding. These findings link that ATP usage in the ER with mitochondrial oxidative phosphorylation and glycolysis. These are very interesting findings and provide important and novel information on ATP metabolism, sources and regulation in the ER. This is an interesting study that significantly advances our knowledge about ER homeostasis and its links to cellular metabolism. It will certainly generate broad interest and further research in this important field of cell biology.

The biochemical/biophysical and cell biological assays demonstrating ATP production and transport are rather extensive and elegant using state-of-the-art techniques and ERTA probe for measurement of ER organellar ATP levels.

However, a few important points remain to be addressed by the authors.

The authors should define what they mean by "Ca^2+^ gradient across the ER membrane". Is there a particular threshold of the gradient? Is this highly regulated process?

The role of cytoplasmic calcium is intriguing. Do the authors have any evidence for a specific signaling event that drives cytosolic calcium-dependent ATP movement? Is *slc35b1* a calcium binding protein? Is a specific calcium gradient required? Could this be a clue to the "memory" of the transporter (subsection “ER ATP import requires a Ca^2+^ gradient across the ER membrane”)?

I wonder why do authors exclusively use FACS analysis for the ERTA probe? Has there been any attempts to carry out a single cell analysis? This would provide additional validation of the data and allow real-time analysis of molecular movements.

The authors should consider using BAPTA-AM (or other cytoplasmic calcium chelators) in life cells to control cytoplasmic calcium in order to complement saponin permeabilized cell analyses.

What is the source of calcium in the cytoplasm that affects ATP transport? Is a store-operated calcium entry involved in addition to the ER?

Is the mitochondria associated membrane involved in the process? Latest report on the role of IRE1 in control of calcium movement between ER and mitochondria (PMID:31110288) indicates a link between UPR, ER, mitochondria and calcium. Is there a link to ATP transport reported here? Can this be documented? This, perhaps, may help to document that ATP and calcium movements are indeed operate or not as two distinct modes.

Reviewer #2

The manuscript by Yong et al. titled "Mitochondria supply ATP to the ER through a mechanism antagonized by cytosolic Ca^2+^" addresses aspects of ER energy homeostasis and its regulation, in particular by cytosolic calcium levels. Yong and colleagues determined the crosstalk between the energy producing mitochondria and the ATP consuming ER lacking an autonomous ATP regenerating machinery. Based on a series of cell-based assays and employing a multitude of fluorescent sensors in combination with all sorts of pharmacological interventions in three cell lines from different species the authors conclude, that

– the ATP supply of the ER is mainly covered by mitochondrial OxPhos

– cytosolic Ca^2+^ regulates the ER ATP import, which is mediated by SLC35B1/AXER

– that induction of protein misfolding in the ER elevates its ATP demand.

Those are indeed interesting findings that further our understanding of ER homeostasis and regulatory mechanisms, granted. However, the manuscript lacks some consistency both experimentally and visually. We would appreciate if the authors could address the following major and minor comments. In the absence of respective revisions, the work does clearly not deserve the rating "outstanding", which is a prerequisite for publication in *eLife*.

Abstract /Significance:

1) The Abstract gives the unwarranted impression that HeLa cells were used throughout the study, i.e. that the so-called calcium antagonized ATP transport into the ER of hamster CHO- and rat INS1-cells was also observed for HeLa cells, which is at odds with Figure 3F of this manuscript (as well as findings by Vishnu et al., 2014, and Klein et al., 2018). Actually, Vishnu et al., 2014 reported a similar phenomenon as for HeLa cells also for INS1-cells. Also see point 23 below.

2) The significance statement gives the unwarranted impression that it was shown for all three cell lines that the calcium antagonized ATP transport into the ER is mediated by SLC35B1/AXER. In fact, only HeLa cells were used for the *slc35b1* knock down experiment. Thus, it remains an open question how calcium antagonized ATP transport into the ER of CHO- and INS1-cells is mediated. After all, Vishnu et al., 2014 as well as Depaoli et al., 2018 – both involving R. Malli as senior author – reported that there are cell type-specific effects involved.

3) The authors emphasize the anti-Warburg effect reflecting an increase in mitochondrial OxPhos and decrease in glycolysis. If I understand correctly, the drop in glycolysis is only shown in Figure 4—figure supplement 1 indirectly by inhibiting OxPhos in stressed and unstressed CHO cells.

– The figure should be a main figure, if its implications deserve mentioning in Abstract and significance statement.

– Is there a better way to address the efficiency of glycolysis in unstressed and stressed cells directly using the Seahorse (e.g. like the ECAR measurements shown in Figure 1—figure supplement 4B)?

Introduction

4) The Introduction is too short and superficial lacking a somewhat more comprehensive background on the topic of ER ATP import. It appears to be somewhat outdated. Maybe a few more references could also be cited.

5) Furthermore, when first talking about ER-ANT it should be pointed out that it is restricted to plants and the respective reference should be given.

6) In addition, the statement that "the mammalian equivalent of ER-ANT1 had remained elusive until a recent publication identified SLC35B1 as the putative mammalian ATP transporter" is misleading, as no orthologs of ER-ANT1 were detected in mammals or yeast and orthologs of SLC35B1 exist in plants and yeast.

Results

7) The Results section needs a substantial re-arrangement. It appears way too crowded and lacks a continuous, smooth flow. For example:

– Why do the authors introduce the inducible H9 CHO model in Figure 1A and 1B to demonstrate the localization of the ERAT probe being independent from ER stress. This should be added to Figure 4 when the authors actually show the increase of F8 production in response to SAHA treatment and the subsequent activation of UPR. The rest of the data for Figure 1 (1C-1H) was generated using the CHO model.

– Figure 1C should be moved to the supplement, its basic knowledge. The "formula" within the ER (ATP supply – ATP consumption = ATP status proportional to ERAT FRET) should be displayed differently.

– Figure 2 starts with INS1 cells and continues with CHO cells. Maybe it`s better to show only the CHO cells and put the data with the INS1 cells reacting similarly as the CHO cells to the supplement. Comparing Figure 2A-D with 2E-H two parameters were changed, cell type and SERCA inhibitor.

8) It would be important to show at least once that BiP activity is compromised under conditions of OxPhos inhibition in CHO- and INS1-cells, i.e. to provide independent biochemical evidence to back up the imaging data, e.g. in connection with Figure 1.

9) Results section: "2-DG administration did not negatively affect the OCR". This is misleading as the 2-DG treatment does affect the OCR positively, which the authors later write. The corresponding red 2-DG lines in Figure 1D and Figure 1—figure supplement 4B lack the standard error for a meaningful comparison to the control.

10) Figure 1E and 1F do not correlate well with regard to the value numbers and tiny error bars in 1F. Figure 1E shows no error bar. What would be the 0 min start value in Figure 1E? Is it the same for the populations?

11) Results section and Figure 1E: "flow-cytometry circumvents photo-bleaching"; But in Figure 1E the DMSO control shows a drop in ER ATP levels as pronounced as the OxPhos inhibitors over time. Is that photo-bleaching or what effect causes the drop in ATP levels over the 30 min time frame of the measurement?

12) Why was the vehicle control DMSO used in 0.01% (Figure 1—figure supplement 5) and 1% (Figure 1—figure supplement 6). The latter is way too high to ensure measurements under physiological conditions.

13) Results subsection “ER ATP comes from Mitochondrial OxPhos in CHO cells” paragraph three and Figure 1G: IAA increases ER ATP levels, but in Figure 1—figure supplement 6B 2-DG reduces ER ATP levels. Can the 2-DG treatment, which is probably a more specific glycolysis inhibitor than IAA, be done with the flow cytometry assay as well? In fact, Iodoacetamide should not be considered as a specific inhibitor of glyceraldehyde phosphate dehydrogenase (GAPDH), it alkylates all cysteine residues.

14) Results section: "Finally, the observations made in H9 CHO cells were completely reproduced in CHO cells that do not express human F8"

This is unclear to me and leads to several questions.

– Are all previous measurements performed in "stressed" CHO cells?!

– Why is the basal ER ATP level (FRET ratio) in Figure 1—figure supplement 7D for DMSO much higher than other measurements (e.g. Figure 1E, 2E, 2E S2C). The stressed cells should not be considered physiological measurements under steady state.

– In Figure 1—figure supplement 7D for DMSO the measurement is constant over time in contrast to Figure 1E and many other vehicle control measurements. Why?

15) Why is the 0 min time point so often omitted in figures?

16) Figure 2: H9 CHO cells with or without induction of F8?

17) It is often unclear how many cells were used in the fluorescence ratiometric analysis/ measurements?

18) Subsection “Cytosolic Ca^2+^ inhibits ATP import into the ER”: "ER ATP usage arrest"

How would SERCA inhibition cause an ATP usage arrest? Wouldn't misfolding or ER stress drive ATP consumption by chaperones?

19) Figure 3H: Writing too small and figure too crowded/complicated. Reduce irrelevant information.

20) Figure 4A: Too small

Would it not be better, having the data from Figure 1A and 1B together with the Western blot panel that demonstrates F8 synthesis upon SAHA treatment and the inductions of UPR?

21) Subsection “Protein misfolding in the ER increases both ER ATP dependence on OxPhos and Tg-triggered Ca^2+^ mobilization into mitochondria.”: it is unclear how long the cells were pre-treated with SAHA for stress induction.

22) Figure 4 I and J: Highlight the trace of population mean.

Discussion

23) Subsection “Cytosolic Ca^2+^ regulates ATP import into the ER” and others: "ER Ca^2+^ release with a transient ER ATP increase (Vishnu et al., 2014) (and repeated in Figure 3F) could reflect a temporary reduction in ER ATP consumption, rather than reflecting the luminal Ca^2+^ sensitivity of the ER ATP import machinery." It appears (e.g. based on Vishnu et al., 2014) that the calcium depletion of the ER slightly precedes the initial increase in ATP level. According to the model by Yong et al. this would already inhibit SLC35B1 and prevent further ATP uptake. Yet, initially ATP levels rise in response to SERCA inhibition. In addition, the authors write "a temporary reduction in ER ATP consumption". How would the reduction of ATP consumption increase the ATP level? Wouldn't it cause a stagnation of the ATP levels without the ATP transporter being active?

24) In general, we wonder how the observed role of cytosolic AMPK in Ca^2+^ coupled ER ATP increase (Vishnu et al., 2014) fits into the model, which is described here, and why this was not addressed or at least discussed here.

---

## [Author Response]

Essential revisions:The major conclusions are of sufficient interest for publication in eLife, as the work addresses important aspects of ER biology. However, there are some issues that need to be addressed before the paper can be recommended for publication in eLife.

The wise and encouraging comments by the editors are greatly appreciated.

We concur with Drs. Sonenberg and Pfeffer that regulation governing ATP supply to the ER constitutes the fundamental aspect of ER biology, esp. for energy homeostasis. Our discovery that cytosolic Ca^2+^ playing such a role witnesses, again, the elegant balancing mechanism of the intricate homeostatic control, inside a mammalian cell.

1) The link between Slc35B1 and CaATiER is weak. The data on HeLa cells do not support the CaATiER model and there are no data on SLC35B1/AXER for the other two cell lines. Therefore, additional work is required to provide this link or, alternatively, to identify the ATP carrier that is linked to CaATiER in INS1 or CHO cells.

First, while SLC35B1 is a necessary component for ATP supply to the ER, as demonstrated by Klein et al., we believe it is not sufficient to explain the *CaATiER* phenomenon. By analogy, for the plasma membrane ATP-sensitive K^+^_ATP_ channel, the ATP-sensing subunits SUR1/SUR2A/SUR2B are separate entities from the K^+^ channel (encoded by the KCNJ-8/-11 genes).

In this revision, we now demonstrate that *Slc35b1* knockdown attenuated the *CaATiER* phenomenon in both CHO and INS1 cells.

To elaborate:

1) Using the semi-intact CHO cell system, we confirmed that *Slc35b1* knockdown reduced basal ER ATP levels, and attenuated *CaATiER*. The figure is now available as main Figure 3E, and the more detailed results are also provided in Figure 3 —figure supplement 2.

2) We also demonstrated that *Slc35b1* knockdown reduced basal ER ATP levels and attenuated *CaATiER* using single cell FRET imaging. The figures are now displayed as Figure 3 —figure supplement 3 (A to F).

These observations further support our model in Figure 3H.

2) An in-depth analysis of the role of Slc35B1 is needed to strengthen the study. It is required to show that Slc35B1 is indeed the one and only ATP transporter to the ER. The analysis of a cytoplasmic calcium sensitivity/dependence of the Slc35B1 may provide some useful information.

We believe our data presented in Figure 3—figure supplement 2 and Figure 3—figure supplement 3 fully support the model that SLC35B1 is a necessary component for ATP import into the ER. However, as important as SLC35B1 is for ER ATP supply, we cannot state that it is the “one and only ATP transporter into the ER”. As in a recent review (Depaoli et al., 2019), we believe that additional pathways might supply the ER with ATP. See Figure 3 in Depaoli et al., 2019.

Among the above proposed mechanisms, based on our results, we believe the SLC35B1 ATP/ADP translocase constitutes the major ATP import mechanism to the ER, while the retrograde import from the Golgi apparatus constitutes another important ATP supply. The latter may explain why *Slc35b1* knockdown is not lethal for the cell lines we have tested so far. In addition, we are unsure if the translocon channels or non-specific transporters (shaded in blue and green) play any role for meaningful ATP import into the ER.

To analyze the cytoplasmic calcium sensitivity/dependence of *Slc35B1*, we performed Ca^2+^ dose response using the semi-intact H9 CHO cell system which showed that Ca^2+^ sensitivity is ~ 0.5 – 2 µM for the *CaATiER* mechanism in mammalian cells. This result is now provided as Figure 3—figure supplement 5.

In addition, importantly, the nice paper by Klein et al. (Klein et al., 2018) elegantly demonstrated that SLC35B1 expression alone did not confer the Ca^2+^ responsiveness (when ectopically expressed in *E. coli*, c.f. Table 1 by Klein et al., cited below), which suggests that the SLC35B1 protein is the ATP/ADP translocation channel, but does not encode the Ca^2+^ responsiveness element. However, in the absence of data, we refrain from further speculation.

3) The discrepancies between the old and new results in INS1 cells need to be explained.

We do not feel that the “old” and new results are different from each other. Please note that in the study of Vishnu et al., 2014, the time frame of BHQ treatment of INS-1 cells was rather short (~ 3 minutes, in the left panel from Figure S3A of Vishnu et al.) prior to the depletion of ER Ca^2+^ by the administration of the IP3 generating agonists CCh and ATP. In the current study we investigated and followed the BHQ effect on ER ATP content for more than 30 minutes, and still observed an increase in ER ATP at the start of the BHQ treatment, shown in Figure 3—figure supplement 3A.

These criticisms of the work are listed in great detail in the reviews, which are attached in full. In addition, the reviewers offered many thoughtful and constructive suggestions to improve the manuscript.Reviewer #1[…] The authors should define what they mean by "Ca^2+^ gradient across the ER membrane". Is there a particular threshold of the gradient? Is this highly regulated process?

The “Ca^2+^ gradient across the ER membrane” means an uneven distribution of Ca^2+^ cation on the two sides of the ER membrane, where on the cytosolic Ca^2+^ level is low (~ 100 nM) while in the ER lumen Ca^2+^ is high (~ 1 mM). It is so-named by analogy to the “proton gradient” across the mitochondrial inner membrane.

As for the excellent point to determine the “Ca^2+^ gradient” threshold, we demonstrated experimentally, using the semi-intact CHO H9 cell system, that the cytosolic Ca^2+^ concentration needed to trigger *CaATiER* is within the high nM to low µM range of free Ca^2+^, as shown in our response to Q2 above. Unfortunately, it is technically challenging to decide such thresholds in an intact cell.

The role of cytoplasmic calcium is intriguing. Do the authors have any evidence for a specific signaling event that drives cytosolic calcium-dependent ATP movement? Is slc35b1 a calcium binding protein? Is a specific calcium gradient required? Could this be a clue to the "memory" of the transporter (subsection “ER ATP import requires a Ca^2+^ gradient across the ER membrane”)?

We did not set out to determine whether the *CaATiER* mechanism is mediated by the Ca^2+^ responsiveness of SLC35B1/AXER. In fact, Klein et al. (Klein et al., 2018) suggested that SLC35B1 alone did not confer Ca^2+^ responsiveness when ectopically expressed in *E. coli* (Table 1 by Klein et al), which suggests that SLC35B1 is solely the ATP/ADP translocation channel.

However, in the absence of data, we refrain from further speculate.

I wonder why do authors exclusively use FACS analysis for the ERTA probe? Has there been any attempts to carry out a single cell analysis? This would provide additional validation of the data and allow real-time analysis of molecular movements.

We relied heavily on FACS as the population statistics is robust and highly reproducible.

In addition, we consider the microscopy-based FRET assays as “single cell analyses”, as suggested by this reviewer, which are shown in Figures 2, 3 and 4. A good example in point is Figure 2 B – D and J. Importantly, the results from both methods are highly consistent.

The authors should consider using BAPTA-AM (or other cytoplasmic calcium chelators) in life cells to control cytoplasmic calcium in order to complement saponin permeabilized cell analyses.

As suggested by this reviewer, we now show the effect of BAPTA-AM effect by microscopy-based and flow-based ER ATP assays. The results are summarized below:

A) While the initial Ca^2+^ chelating by BAPTA-AM supported our *CaATiER* model in that ER ATP FRET signal went up transiently, we caution the conclusion that BAPTA-AM is an *intracellular* Ca^2+^ chelator and therefore the result cannot be interpreted as a cytoplasmic Ca^2+^ chelating effect.

B) Prolonged BAPTA-AM treatment (5 µM x 90min) significantly reduced ER ATP, as detected by flow-cytometry, shown in Author response image 1.

What is the source of calcium in the cytoplasm that affects ATP transport? Is a store-operated calcium entry involved in addition to the ER?

From results in Figure 2F, and Figure 2J, it appears that Ca^2+^ release from the ER pool is sufficient, as extracellular chelation by EGTA did not abolish the *CaATiER* phenomenon.

We conclude that store-operated calcium entry (SOCE) is not required based on the current data. Whether SOCE is involved in an intact cell under physiological conditions is an excellent question for future investigation.

Is the mitochondria associated membrane involved in the process? Latest report on the role of IRE1 in control of calcium movement between ER and mitochondria (PMID:31110288) indicates a link between UPR, ER, mitochondria and calcium. Is there a link to ATP transport reported here? Can this be documented? This, perhaps, may help to document that ATP and calcium movements are indeed operate or not as two distinct modes.

This is another excellent point from this reviewer. The topic of “mitochondria-associated membrane” is a broad field to address. We have now performed the following experiment, using established tools, as suggested by this reviewer.

By applying the artificial ER-to-mito linkers characterized by Csordas et al. (Csordas et al., 2010), we failed to detect any meaningful changes in the oxygen consumption rates (OCR) after linker ligation by Rapamycin (data in Author response image 2 in panel B after “Rapa”). Furthermore, unfortunately, the design of these fluorescence linkers (CFP-FRB-ER and RFP-FKBP-mito, respectively) prevented our ER ATP analysis using our ERAT protein that employs the same CFP as the FRET donor. Nevertheless, we agree with this reviewer that his/her question is an important direction for future investigation.

**Author response image 2. respfig2:** Left panel – Illustration of linkers used; Right panel – OCR results before and after linkers ligation by Rapa. DT-8 = Double transfected clone 8, with both linkers present, experimental group. AR-6/-10* = AKAP1-mRFP, with only mito-linker present, clone-6/-10. *AR clones served as control clones.

Reviewer #2[…] Those are indeed interesting findings that further our understanding of ER homeostasis and regulatory mechanisms, granted. However, the manuscript lacks some consistency both experimentally and visually. We would appreciate if the authors could address the following major and minor comments. In the absence of respective revisions, the work does clearly not deserve the rating "outstanding", which is a prerequisite for publication in eLife.

We thank reviewer #2 for his positive comments. We would like to clarify the second conclusion by reviewer #2, i.e. “cytosolic Ca^2+^ regulates the ER ATP import, which is mediated by SLC35B1/AXER”. While Klein et al. demonstrated (Klein et al., 2018) SLC35B1/AXER mediates ER ATP import, and in this revision we provide analysis in CHO, HeLa and INS-1 cells that SLC35B1 is indeed indispensable for ER ATP import. We do not claim that the Ca^2+^-responsiveness is also conferred by SLC35B1, as we elaborated above in Q2 to the editors.

To avoid this confusion, we modified our working hypothesis cartoon to isolate Ca^2+^-sensing element as an independent entity (now as Figure 3H). In the absence of data, we refrain from further speculation on the Ca^2+^-responsiveness of SLC35B1.

Abstract /Significance:1) The Abstract gives the unwarranted impression that HeLa cells were used throughout the study, i.e. that the so-called calcium antagonized ATP transport into the ER of hamster CHO- and rat INS1-cells was also observed for HeLa cells, which is at odds with Figure 3F of this manuscript (as well as findings by Vishnu et al., 2014, and Klein et al., 2018). Actually, Vishnu et al., 2014 reported a similar phenomenon as for HeLa cells also for INS1-cells. Also see point 23 below.

We thank Dr. Zimmermann for this comment.

By applying the semi-intact CHO cell assay, which was not available in 2014, we are now confident that cytosolic Ca^2+^ plays an inhibitory role on ER ATP import. The result is now available as Figure 3E.

In addition, microscopy-based imaging also shows that cytosolic Ca^2+^ attenuates ER ATP import in all three cell lines, CHO/HeLa/INS-1. The results are presented in Figure 3—figure supplement 3, and also displayed above in response to Q2 of the editors.

In addition, in the original publication Vishnu et al. cautiously concluded “the Ca^2+^-coupled ER ATP increase is independent of the mode of Ca^2+^ mobilization and controlled by the rate of ATP biosynthesis” (Vishnu et al., 2014). Nevertheless, while we confirmed that the initial ER ATP increase is highly reproducible in HeLa (<10 min in our Figure 3F, as well as in Vishnu et al., 2014 and Klein et al., 2018) and INS-1 cells, it does not represent the long-term ER ATP change in response to SERCA inhibition by BHQ. We are pleased to have faithfully recorded the dynamic changes in ER ATP, which shows cell-type dependent variation as suggested by the reviewer.

2) The significance statement gives the unwarranted impression that it was shown for all three cell lines that the calcium antagonized ATP transport into the ER is mediated by SLC35B1/AXER. In fact, only HeLa cells were used for the slc35b1 knock down experiment. Thus, it remains an open question how calcium antagonized ATP transport into the ER of CHO- and INS1-cells is mediated. After all, Vishnu et al., 2014 as well as Depaoli et al., 2018 – both involving R. Malli as senior author – reported that there are cell type-specific effects involved.

We agree with Dr. Zimmermann that there are cell-type specific effects involved immediately upon SERCA inhibition. Now with the *CaATiER* phenomenon confirmed in all three cell lines, we conclude:

1) The *CaATiER* phenomenon is not cell type-dependent, as shown in Figure 3F (HeLa cell) and Figure 3—figure supplement 3 (A and D, INS-1 and CHO H9 cells, respectively).

2) There is a short-term transient increase in ER ATP after SERCA inhibition which may result from arrest of ER ATP usage. However, the long-term effects on ER ATP depletion are consistent in all three cell lines as shown in Figure 3F and Figure 3—figure supplement 3 A and D.

3) Intracellular Ca^2+^ trafficking is a complex process in itself, and we are able to confirm the *CaATiER* phenomenon using semi-intact CHO cells where we confirmed the cytosolic Ca^2+^ effect is inhibitory for ER ATP import, as demonstrated in Q2 response to the editors.

3) The authors emphasize the anti-Warburg effect reflecting an increase in mitochondrial OxPhos and decrease in glycolysis. If I understand correctly, the drop in glycolysis is only shown in Figure 4—figure supplement 1 indirectly by inhibiting OxPhos in stressed and unstressed CHO cells.– The figure should be a main figure, if its implications deserve mentioning in Abstract and significance statement.

We are confident about the “anti-Warburg effect” in CHO H9 cells.

However, we refrain from further speculation by considering other aspects of cell metabolism, esp. glucose metabolism.

– Is there a better way to address the efficiency of glycolysis in unstressed and stressed cells directly using the Seahorse (e.g. like the ECAR measurements shown in Figure 1—figure supplement 4B)?

ECAR is not an optimal measure for glycolysis, esp. for CHO cells with a vigorous TCA cycle that produces organic acids. In order to support our claim that glycolysis is reduced in stressed cells, we performed flow-cytometry using 2-NBDG labeling to measure glucose uptake, upon FVIII induction with HDAC inhibitors, NaB or SAHA. The results are consistent with our hypothesis that ER stress reduces glucose uptake in CHO H9 cells and are now shown as Figure 4—figure supplement 2.

Introduction4) The Introduction is too short and superficial lacking a somewhat more comprehensive background on the topic of ER ATP import. It appears to be somewhat outdated. Maybe a few more references could also be cited.

Introduction” was modified as suggested.

5) Furthermore, when first talking about ER-ANT it should be pointed out that it is restricted to plants and the respective reference should be given.

Modified as suggested.

6) In addition, the statement that "the mammalian equivalent of ER-ANT1 had remained elusive until a recent publication identified SLC35B1 as the putative mammalian ATP transporter" is misleading, as no orthologs of ER-ANT1 were detected in mammals or yeast and orthologs of SLC35B1 exist in plants and yeast.

Modified as suggested.

Results7) The Results section needs a substantial re-arrangement. It appears way too crowded and lacks a continuous, smooth flow. For example:– Why do the authors introduce the inducible H9 CHO model in Figure 1A and 1B to demonstrate the localization of the ERAT probe being independent from ER stress. This should be added to Figure 4 when the authors actually show the increase of F8 production in response to SAHA treatment and the subsequent activation of UPR. The rest of the data for Figure 1 (1C-1H) was generated using the CHO model.

We apologize for the confusion, but the data for Figure 1 C to H panels were generated using the H9 CHO cells. Specifically, H9 CHO is an inducible cell system for FVIII production which has no UPR activation under basal conditions, as we demonstrated previously (Dorner et al., 1989) and here in *Figure 4A*.

To emphasize, the only experiment that used parental CHO cells (i.e. DUK clone) is in Figure 1—figure supplement 7D.

– Figure 1C should be moved to the supplement, its basic knowledge. The "formula" within the ER (ATP supply – ATP consumption = ATP status proportional to ERAT FRET) should be displayed differently.

We thank Dr. Zimmermann for this suggestion. However, it appeared to us that not all readers may be familiar about ER ATP homeostasis so we have not moved this figure.

– Figure 2 starts with INS1 cells and continues with CHO cells. Maybe it`s better to show only the CHO cells and put the data with the INS1 cells reacting similarly as the CHO cells to the supplement. Comparing Figure 2A-D with 2E-H two parameters were changed, cell type and SERCA inhibitor.

We thank Dr. Zimmermann for this suggestion. However, we strongly believe *CaATiER* is better supported by data from two cell lines, esp. from rat and Chinese hamster respectively, using two SERCA inhibitors that are structurally distinct. We were convinced by this comparison *CaATiER* is not a compound or cell type-specific artifact. Also, BHQ is selected for INS-1 cells due to its reversibility (Figure 2—figure supplement 1), while Tg is a highly efficient but irreversible SERCA inhibitor.

8) It would be important to show at least once that BiP activity is compromised under conditions of OxPhos inhibition in CHO- and INS1-cells, i.e. to provide independent biochemical evidence to back up the imaging data, e.g. in connection with Figure 1.

We thank Dr. Zimmermann for this suggestion. The data highly consistent with our findings were shown in an earlier paper by Preissler et al., which we are citing for BiP trans-protomer formation in response to Tg.

In the above referenced publication, BiP oligomerization was proposed as a consequence from ATP depletion by Preissler et al., although in this reference the authors did not realize the ER ATP levels are controlled by cytosolic Ca^2+^.

However, we are also aware that BiP activity is not solely determined by ER ATP abundance, such as ERdj proteins, AMPylation and Ca^2+^, and is an inferior marker to our ERAT probe for determining ER ATP levels. For detailed discussions on this topic, please refer to relevant publications by Drs. Hendershot and Ron (Amin-Wetzel et al., 2017; Preissler et al., 2015; Wei et al., 1995).

9) Results section: "2-DG administration did not negatively affect the OCR". This is misleading as the 2-DG treatment does affect the OCR positively, which the authors later write. The corresponding red 2-DG lines in Figure 1D and Figure 1—figure supplement 4B lack the standard error for a meaningful comparison to the control.

We agree with Dr. Zimmermann for the observation that 2-DG treatment does affect the OCR positively. However, the difference was not statistically significant as shown in Figure 1—figure supplement 4A, denoted by “n.s.”.

The absence of standard error bars in Figure 1D was due to small error value. Using Prism software we received the message “For some points, the error bars would be shorter than the height of the symbol. In these cases, Prism simply does not draw the error bars.”

10) Figure 1E and 1F do not correlate well with regard to the value numbers and tiny error bars in 1F. Figure 1E shows no error bar. What would be the 0 min start value in Figure 1E? Is it the same for the populations?

Yes. Firstly, the 0min start value are all the same for every sample, as all tubes originated the same stock of reporter CHO cells, due to the experimental design available in our “Transparent reporting form”, cited below for clarification. Specifically, for Figure 1E, all tubes with OxPhos toxins received the same batch of reporter cells from one cell suspension preparation (Step 6 – Step 7), to ensure cell population homogeneity for comparison.

Secondly, flow cytometry-based FRET assay samples thousands of cells per time point.

For example, in Figure 1F, N equals 5388 for DMSO, 5298 for Oligo, 5313 for Rotenone, 5288 for FCCP groups, respectively. For Figure 1E, Prism software generated this message reading – “For some points, the error bars would be shorter than the height of the symbol. In these cases, Prism simply does not draw the error bars.”

**Author response image 3. respfig3:** 

11) Results section and Figure 1E: "flow-cytometry circumvents photo-bleaching"; But in Figure 1E the DMSO control shows a drop in ER ATP levels as pronounced as the OxPhos inhibitors over time. Is that photo-bleaching or what effect causes the drop in ATP levels over the 30 min time frame of the measurement?

Firstly, the basal level drop is not from photobleaching, as every cell is only sampled once in a flow-cytometer.

Secondly, we think the baseline drift was derived from the vigorous shaking before cells were taken up by the flow cytometer, and we are working diligently to modify our procedures to mitigate such an effect, by reducing the shaking speed. The improvement is visible in Figure 2E and 2F. However, shaking step cannot be omitted for flow procedure.

12) Why was the vehicle control DMSO used in 0.01% (Figure 1—figure supplement 5) and 1% (Figure 1—figure supplement 6). The latter is way too high to ensure measurements under physiological conditions.

The DMSO concentration was used to match the toxin dilution, in which the stock was prepared in. In addition, incubation of cell in 1% (vol/vol) DMSO over the period for 1 hr did not affect cell physiology.

13) Results subsection “ER ATP comes from Mitochondrial OxPhos in CHO cells” paragraph three and Figure 1G: IAA increases ER ATP levels, but in Figure 1—figure supplement 6B 2-DG reduces ER ATP levels. Can the 2-DG treatment, which is probably a more specific glycolysis inhibitor than IAA, be done with the flow cytometry assay as well? In fact, Iodoacetamide should not be considered as a specific inhibitor of glyceraldehyde phosphate dehydrogenase (GAPDH), it alkylates all cysteine residues.

2-DG, if used for 2hrs, effectively depleted total cellular ATP (Figure 1—figure supplement 8B), due to its metabolism by hexokinase. The more imminent effect of 2-DG was tested and displayed in Figure 1—figure supplement 7.

We agree with Dr. Zimmermann that there is no single “glycolysis inhibitor” that only blocks ATP regeneration, as we discussed in our Introduction. For this reason, we used “no glucose” condition (Figure 1H, “Glucose” vs. “PBS”) to confirm the effect of blocking glycolysis, where glucose addition for one hour had no effect.

14) Results section: "Finally, the observations made in H9 CHO cells were completely reproduced in CHO cells that do not express human F8"This is unclear to me and leads to several questions.– Are all previous measurements performed in "stressed" CHO cells?!

Please refer to our response to comment 7 for clarification.

H9 CHO is not considered “stressed” by basal level of F8 expression (Figure 4A), which is under the ER proteostasis control. We emphasize that only under prolonged treatment with a chemical inducer of FVIII transcription (HDAC inhibitor, e.g. SAHA or NaB) are the H9 CHO cells “stressed” (Figure 4A).

– Why is the basal ER ATP level (FRET ratio) in Figure 1—figure supplement 7D for DMSO much higher than other measurements (e.g. Figure 1E, 2E, 2E S2C). The stressed cells should not be considered physiological measurements under steady state.

The above explanation assures the basal H9 CHO cells are not “stressed”.

In Figure 1—figure supplement 7D, CHO cells without F8 expression were used. The baseline FRET level is affected by the probe ERAT expression level, which is demonstrated in Figure 1—figure supplement 3A. Therefore, unlike the ATP luminescence assay by ATPLite kit, the baseline FRET signal cannot be used to calibrate ER ATP levels, and the difference in baseline FRET ratios are considered as clonal variation between the H9 and CHO cell clones. The change in FRET signal (*ΔFRET*) is more meaningful by design (Imamura et al., 2009).

– In Figure 1—figure supplement 7D for DMSO the measurement is constant over time in contrast to Figure 1E and many other vehicle control measurements. Why?

The CHO cells without F8 expression (i.e. DUK cells) use much less ATP in the ER, which correlates with less OxPhos activity in these cells. This observation is consistent with our notion that ER stress leads to an “*anti-Warburg*” effect.

15) Why is the 0 min time point so often omitted in figures?

We provided an explanation in the response to comment 10. The 0 min cell population homogeneity was ensured by experimental design.

16) Figure 2: H9 CHO cells with or without induction of F8?

H9 CHO cells used throughout Figures 1 to 3 were not induced for F8 expression, therefore they were not stressed.

17) It is often unclear how many cells were used in the fluorescence ratiometric analysis/ measurements?

We provided the information in the “Transparent Reporting file”, with the uploaded data files contained all the information needed for the statistical analysis.

We re-emphasize here that more than 2900 cells were used per data point for flow-analysis. We thought that would not contribute significantly to the message.

18) Subsection “Cytosolic Ca^2+^ inhibits ATP import into the ER”: "ER ATP usage arrest"How would SERCA inhibition cause an ATP usage arrest? Wouldn't misfolding or ER stress drive ATP consumption by chaperones?

It was our hypothesis for “ER ATP usage arrest”, as many Ca^2+^-dependent ER chaperones also promote ATP hydrolysis, such as Calnexin and Calreticulin in cooperation with BiP.

Yes, protein misfolding should drive ATP consumption by chaperones. This partially explained why ER ATP drops in H9 cells upon matrix detachment and shaking, a condition that hampers OxPhos inside a cell (Grassian et al., 2011). However, ER Ca^2+^ depletion is not to be taken as equivalent to ER stress/protein misfolding induction, as shown recently (Szalai et al., 2018).

19) Figure 3H: Writing too small and figure too crowded/complicated. Reduce irrelevant information.

We apologize for the methodology description. We modified our *CaATiER* model as suggested.

20) Figure 4A: Too smallWould it not be better, having the data from Figure 1A and 1B together with the Western blot panel that demonstrates F8 synthesis upon SAHA treatment and the inductions of UPR?

We apologize for the small font, as it is a repeat and confirmation of our previous observation (Dorner et al., 1989). We chose not to introduce WB result from SAHA treatment to avoid confusion between basal “unstressed” H9 CHO and the “stressed” H9 CHO.

21) Subsection “Protein misfolding in the ER increases both ER ATP dependence on OxPhos and Tg-triggered Ca^2+^ mobilization into mitochondria.”: it is unclear how long the cells were pre-treated with SAHA for stress induction.

We apologize for the omission. The SAHA treatment was 21 hrs for Figure 4 C and D, and this information was inserted in the figure legend.

22) Figure 4 I and J: Highlight the trace of population mean.

Highlighted.

Discussion23) Subsection “Cytosolic Ca^2+^ regulates ATP import into the ER” and others: "ER Ca^2+^ release with a transient ER ATP increase (Vishnu et al., 2014) (and repeated in Figure 3F) could reflect a temporary reduction in ER ATP consumption, rather than reflecting the luminal Ca^2+^ sensitivity of the ER ATP import machinery." It appears (e.g. based on Vishnu et al., 2014) that the calcium depletion of the ER slightly precedes the initial increase in ATP level. According to the model by Yong et al. this would already inhibit SLC35B1 and prevent further ATP uptake. Yet, initially ATP levels rise in response to SERCA inhibition. In addition, the authors write "a temporary reduction in ER ATP consumption". How would the reduction of ATP consumption increase the ATP level? Wouldn't it cause a stagnation of the ATP levels without the ATP transporter being active?

We acknowledge that we do not completely understand the initial transient increase in the ERAT FRET signal in HeLa cells, except that it is highly reproducible in certain cell types (to a lesser extent INS-1, as shown in Figure 3—figure supplement 3A). We suspect there is a lag time in the cytosolic Ca^2+^ elevation to induce the *CaATiER* machinery, in the sequence of events in temporal order: SERCA inhibitor -> ER Ca^2+^ uptake arrest -> ER Ca^2+^ release -> cytosolic Ca^2+^ accumulation -> *CaATiER* activation.

Considering our semi-intact CHO cell result regarding the dose-response of [Ca^2+]^(Figure 3—figure supplement 5), we inferred that ~1 µM cytosolic Ca^2+^ isthe activation threshold, therefore it is reasonable to assume that *CaATiER* mechanism activation lagged behind the SERCA inhibitor administration.

24) In general, we wonder how the observed role of cytosolic AMPK in Ca^2+^ coupled ER ATP increase (Vishnu et al., 2014) fits into the model, which is described here, and why this was not addressed or at least discussed here.

Wedid not further investigate the AMPK phosphorylation status here, partly in that we have more direct measure of ATP. We added discussions on the possible role of AMPK, in subsections “Cytosolic Ca2+ attenuates ATP import into the ER” and “ER protein misfolding increases the ER bioenergetic requirement for ATP from OxPhos”.

References

Amin-Wetzel, N., Saunders, R.A., Kamphuis, M.J., Rato, C., Preissler, S., Harding, H.P., and Ron, D. (2017). A J-Protein Co-chaperone Recruits BiP to Monomerize IRE1 and Repress the Unfolded Protein Response. Cell 171, 1625-1637 e1613.

Csordás, G., Várnai, P., Golenár, T., Roy, S., Purkins, G., Schneider, T.G., Balla, T., and Hajnóczky, G. (2010). Imaging Interorganelle Contacts and Local Calcium Dynamics at the ER-Mitochondrial Interface. Mol Cell 39, 121-132.

Depaoli, M.R., Hay, J.C., Graier, W.F., and Malli, R. (2019). The enigmatic ATP supply of the endoplasmic reticulum. Biol Rev Camb Philos Soc *94*, 610-628.

Dorner, A.J., Wasley, L.C., and Kaufman, R.J. (1989). Increased synthesis of secreted proteins induces expression of glucose-regulated proteins in butyrate-treated Chinese hamster ovary cells. J Biol Chem *264*, 20602-20607.

Preissler, S., Rato, C., Chen, R., Antrobus, R., Ding, S., Fearnley, I.M., and Ron, D. (2015). AMPylation matches BiP activity to client protein load in the endoplasmic reticulum. *ELife* 4, e12621.

Szalai, P., Parys, J.B., Bultynck, G., Christensen, S.B., Nissen, P., Moller, J.V., and Engedal, N. (2018). Nonlinear relationship between ER Ca(2+) depletion versus induction of the unfolded protein response, autophagy inhibition, and cell death. Cell Calcium *76*, 48-61